# Diff-MN: Diffusion Parameterized MoE-NCDE for Continuous Time Series Generation with Irregular Observations

Xu Zhang [1]   Junwei Deng [2]   Chang Xu [1]   Hao Li [3]   Jiang Bian [1]

## Abstract

Time series generation (TSG) is widely used across domains, yet most existing methods assume regular sampling and fixed output resolutions. These assumptions are often violated in practice, where observations are irregular and sparse, while downstream applications require continuous and high-resolution TS. Although Neural Controlled Differential Equation (NCDE) is promising for modeling irregular TS, it is constrained by a single dynamics function, tightly coupled optimization, and limited ability to adapt learned dynamics to newly generated samples from the generative model. We propose Diff-MN, a continuous TSG framework that enhances NCDE with a Mixture-of-Experts (MoE) dynamics function and a decoupled architectural design for dynamics-focused training. To further enable NCDE to generalize to newly generated samples, Diff-MN employs a diffusion model to parameterize the NCDE temporal dynamics parameters (MoE weights), i.e., jointly learn the distribution of TS data and MoE weights. This design allows sample-specific NCDE parameters to be generated for continuous TS generation. Experiments on ten public and synthetic datasets demonstrate that Diff-MN consistently outperforms strong baselines on both irregular-to-regular and irregular-to-continuous TSG tasks. The code is available at the link `https://github.com/microsoft/TimeCraft/tree/main/Diff-MN`.

## 1. Introduction

Time series generation (TSG) is essential across many real-world domains, including healthcare (Deng et al., 2025a), finance (Zhang et al., 2025a; Vasiliu et al., 2024; Zhang et al., 2024), manufacturing and energy systems (Zhang et al., 2025b; Fuest et al., 2025; Zhang et al., 2025c). It supports data augmentation (Tang et al., 2025), privacy-preserving data sharing (Tian et al., 2024), and high-fidelity simulation for decision-making (Turowski et al., 2024). Despite substantial progress, most TSG methods assume regularly sampled TS, treating both training and generated data as discrete observations at fixed intervals. As shown in Fig. 1(a), existing approaches typically train on uniformly observed sequences and generate outputs at fixed resolutions (Mushunje et al., 2023; Li, 2023; Huang et al., 2025; Ge et al., 2025). While a few methods allow irregular training data, they still generate sequences at fixed intervals, as illustrated in Fig. 1(b) (Nikitin et al., 2024; Ramponi et al., 2018). This setting does not match real-world scenarios, where observations are often irregular and sparse, yet downstream tasks require continuous and high-resolution TS. For example, in ICU monitoring, electronic health records are irregular and sparse, while applications such as patient trajectory simulation or risk forecasting require dense sequences (Tian et al., 2024; Li et al., 2025a). Consequently, continuous TSG in Fig. 1(c) from irregular observations remains an under-explored but critical challenge.

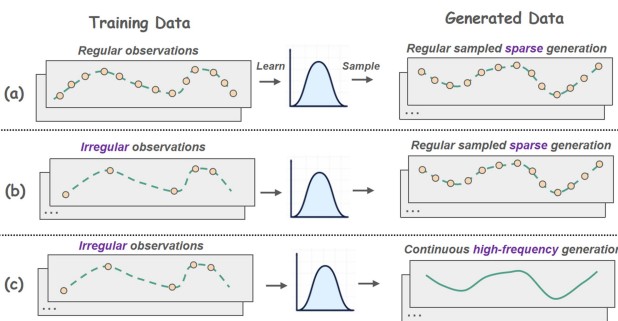

*Figure 1.* Different TSG tasks categorized by the data continuity and sampling irregularity. Unlike (a) and (b), we generate continuous TS from irregular observations, producing high-frequency sequences with richer contextual information.

[1]Microsoft Research, Beijing, China [2]University of Illinois Urbana-Champaign, Illinois, United States [3]Imperial College London, London, United Kingdom. Correspondence to: Chang Xu <chanx@microsoft.com>.

*Proceedings of the 43rd International Conference on Machine Learning*, Seoul, South Korea. PMLR 306, 2026. Copyright 2026 by the author(s).

Continuous-time modeling and irregular observations have been widely studied, leading to several approaches. Among them, Neural Ordinary Differential Equations (NODEs) and their extension, Neural Controlled Differential Equations (NCDE), are particularly popular. By combining continuous-time differential equations with learnable neural networks, NCDE-based methods effectively model temporal dynamics and handle irregular observations. Recent work has applied them to TSG (Lim et al., 2023; Naiman et al., 2024), typically following a two-stage paradigm: irregularly sampled TS are first transformed into regularly sampled ones using NCDE or their variants, and standard TSG models such as GANs (Jeon et al., 2022) and VAEs (Naiman et al., 2024) are then applied to generate data on regular time grids.

However, on the one hand, existing methods mainly use NCDE to process irregular inputs and then generate regularly sampled TS (Fig. 1(b)), with limited exploration of continuous TSG and evaluation (Fig. 1(c)). On the other hand, they directly adopt the standard NCDE without further improvement. Although standard NCDE are effective for modeling irregular TS, they rely on a single dynamics function, lack dynamics-focused optimization design, and cannot generalize learned dynamics to newly generated samples from a generative model. As a result, there remains substantial room to improve standard NCDE for irregular TS modeling and high-quality continuous-time generation.

In this study, we propose Diff-MN, a framework for continuous TSG with irregular observations. Our work differs from existing studies in the following aspects:

1. We are the first to formally study continuous TSG and evaluate it under the generated high-frequency TS data from two perspectives: TS forecasting performance and analytical solutions of cubic polynomials.

2. To enable effective continuous TSG, we improve standard NCDE in two key ways. **(i)** We replace the single dynamics function with a mixture-of-experts (MoE) dynamics function, where different experts capture distinct temporal patterns and are aggregated to model complex dynamics. The learned MoE weights explicitly encode temporal dynamics, providing a basis for using the diffusion model to parameterize them, thereby generalizing to newly generated samples. **(ii)** We propose a dynamics-focused, decoupled optimization strategy for NCDE. Specifically, we replace the State Initialization Network (SIN) and Readout Network (RN) with a pre-trained channel-wise autoencoder, which warm-starts training and allows the model to focus on optimizing the dynamics function.

3. Based on **(i)** and **(ii)**, the MoE weights learned by the trained NCDE encode temporal dynamics that are specific to the training samples. When new samples are generated, their temporal dynamics may differ, leading to a mismatch with these sample-specific weights. To address this, we propose to parameterize the MoE weights for each new sample.

Specifically, we jointly model TS data and their corresponding MoE weights with a diffusion model to learn their joint distribution. As a result, for each newly generated sample, matched MoE weights that reflect the true dynamics are generated and substituted back into the pre-trained NCDE, enabling more accurate continuous time series generation.

Our contributions are as follows:

- We are the first to formally study continuous TSG and evaluate it on generated high-frequency time series, filling an important research gap.

- For better continuous TSG, we first propose a **decoupled Mixture-of-Experts Neural CDE (MoE-NCDE)**, which enhances standard NCDE with the MoE dynamic functions and dynamics-focused optimization design. This enables NCDE to better capture complex and dynamically changing temporal patterns from irregular observations.

- We further propose **Diff-MN**, a continuous TSG framework that jointly models TS data and MoE-NCDE temporal dynamics parameters (MoE weights) using a diffusion model. This design parameterizes the MoE weights and allows sample-specific weights to be generated and substituted back into a pre-trained MoE-NCDE, enabling accurate continuous TSG for newly generated samples.

- We conduct extensive experiments on four popular time-series generation datasets, four medical ECG datasets, and two synthetic datasets. Both quantitative and qualitative results demonstrate that Diff-MN consistently outperforms strong baselines on both irregular-to-regular and irregular-to-continuous generation tasks.

## 2. Related Work

Although limited work directly addresses continuous time series generation, several studies tackle closely related irregular time series modeling problems. Early work extends RNN-based models with irregular time-interval awareness: GRU-D (Che et al., 2018) introduces learnable temporal decay, ODE-RNN (Rubanova et al., 2019) evolves hidden states via ODEs between observations, and Latent ODE models continuous latent trajectories for interpolation and extrapolation at arbitrary time points. Building on these foundations, NCDE-based generators such as KOVAE (Jeon et al., 2022) and GTGAN (Naiman et al., 2024) conduct irregular-to-regular generation using standard NCDEs.

Attention-based methods naturally handle dependencies across arbitrary time steps without assuming regular intervals. HeTVAE (Shukla & Marlin) incorporates continuous-time embeddings into attention, and TrajGPT (Song et al., 2025) introduces a Selective Recurrent Attention mechanism with learnable time decay to support extrapolation to

unseen timestamps. ProFITi (Yalavarthi et al., 2025) instead leverages Conditional Normalizing Flows to enable uncertainty-aware forecasting on irregularly sampled data.

Unlike existing methods that primarily address irregular-to-regular modeling, we formally study the continuous TSG task and propose Diff-MN, a diffusion-based framework built upon a novel decoupled MoE-NCDE architecture. Related work about TSG models for regular data is provided in the Appendix A.

## 3. Method

**Overview.** Our method is shown in Figure 2. We next detail the training and generation process of Diff-MN.

**Training:** First, we train the decoupled MoE NCDE on irregular observations, learning shared State Initialization Network (SIN), Readout Network (RN), and dynamic-function neural networks, along with sample-specific MoE weights that encode each instance's temporal dynamics. These components are then used to impute missing values. Next, to parameterize the MoE weights for generalizing to new samples, we jointly train a diffusion model on the imputed data and their corresponding MoE weights. Consequently, the diffusion model generates new samples together with matched MoE weights that reflect their temporal dynamics, enabling better continuous TSG.

**Generation:** The newly generated samples and MoE weights from the diffusion model are fed back into pretrained MoE-NCDE for continuous TSG, producing high-frequency, longer, and more informative TS.

### 3.1. Preliminary

**Notation.** Suppose we have $N$ observed multivariate TS. Each TS $\mathcal{X}^{(j)} : [0, T] \rightarrow \mathbb{R}^M$, for $j = 1, \ldots, N$, is a continuous multivariate function defined over its time interval, where $M$ is the number of channels/variables. In practice, we observe each multivariate TS at discrete and irregular time points $\{t_0^{(j)}, t_1^{(j)}, \ldots, t_{n_j}^{(j)}\} \subset [0, T]$, where $t_0^{(j)} < t_1^{(j)} < \cdots < t_{n_j}^{(j)}$. The observed dataset is:

$$\mathcal{D} = \left\{ \mathcal{O}^{(j)} = \left\{ (t_i^{(j)}, \mathbf{x}_i^{(j)}, \}_{i=0}^{n_j} \,\middle|\, j = 1, \ldots, N \right\} \quad (1)$$

**Problem Definition.** Given dataset $\mathcal{D}$, the goal is to learn a generative model $p(\mathcal{O})$ over discrete and irregular TS for irregular or continuous generation, i.e., our goal is to learn a model capable of sampling new sequences $\tilde{\mathcal{X}}_{1:T} \sim p_\theta(\mathcal{X}_{1:T})$. The generated dataset is denoted as $\tilde{\mathcal{D}} = \left\{ \tilde{\mathcal{O}}^{(j)} = \left\{ (\tilde{t}^{(j)}, \tilde{\mathbf{x}}^{(j)}, \}^{\tilde{n}_j} \,\middle|\, \tilde{t} \in [0, T] \right\}$ where $\tilde{t}^{(j)}$ denotes arbitrary time step of $j$-th sample.

**Neural Controlled Differential Equation.** Neural CDE (Kidger et al., 2020) extends Neural ODE (Chen et al.,

2018) by modeling latent dynamics driven by a continuous control path. **In continuous TSG, values generated between observations must align with surrounding temporal dynamics to avoid deviating from the true system distribution**. Theoretically, we use NCDE instead of NODE because NCDE conditions the dynamics on irregular observations via a control path, enabling more accurate interpolation, whereas NODE may drift without observation guidance. The details of why NCDE is better in our task **from both theoretical and experimental perspectives can be found in Appendix B.1**. The NCDE is defined as:

$$z(t) = z(0) + \int_0^t f_\theta(z(s)) \, dX(s), \quad (2)$$

where $f_\theta : \mathbb{R}^d \rightarrow \mathbb{R}^d$ is a learnable dynamic-function neural network, and $X(s) : [0, T] \rightarrow \mathbb{R}$ is a continuous interpolation (e.g., a natural cubic spline) of the discrete observations $\{(t_i, x_i)\}_{i=0}^n$. $z(0) \in \mathbb{R}^d$ is the initial latent state mapped from $\mathcal{X}_0$ based on the network SIN.

Notably, although the control path $X(s)$ is smooth due to spline interpolation, the latent trajectory $z(t)$ can still be non-smooth because its evolution is governed by the nonlinear neural network $f_\theta(z(s))$ and the mixture-of-experts, which together allow expressive, non-smooth, and highly diverse temporal patterns. **The proof from both theoretical and experimental perspectives can be found in Appendix B.2**.

Finally, `CDESolver` to solve NCDE is formalized as:

$$\texttt{CDESolver}(f_\theta, z(0), [0, T], X) \rightarrow \{z(t)\}_{t \in [0,T]}, \quad (3)$$

Its output is the latent state $z(t)$ at arbitrary time points $t \in [0, T]$. Then, the network RN subsequently maps this hidden representation back to the original channel space to produce the predicted TS values.

### 3.2. Modeling Diverse Dynamics with Dense MoE

Real-world TS exhibit diverse temporal patterns that a single dynamic function often struggles to capture. Using a Mixture of Experts (MoE), the complex temporal dynamics are first decomposed, allowing each expert to focus on the patterns it is best suited to learn, such as local or global trends. This makes the overall dynamics easier to learn while enhancing both expressiveness and generalization. By combining their outputs, the MoE-NCDE produces richer and more flexible temporal representations. This is strongly supported by Figure 6 in the experimental section.

Notably, we use a dense MoE rather than a sparse one, prioritizing stability over efficiency. Dense MoE fully engages all experts at each step, enabling stable training and better learning of complex temporal dynamics in our task, which involves limited training data and low-parameter models.

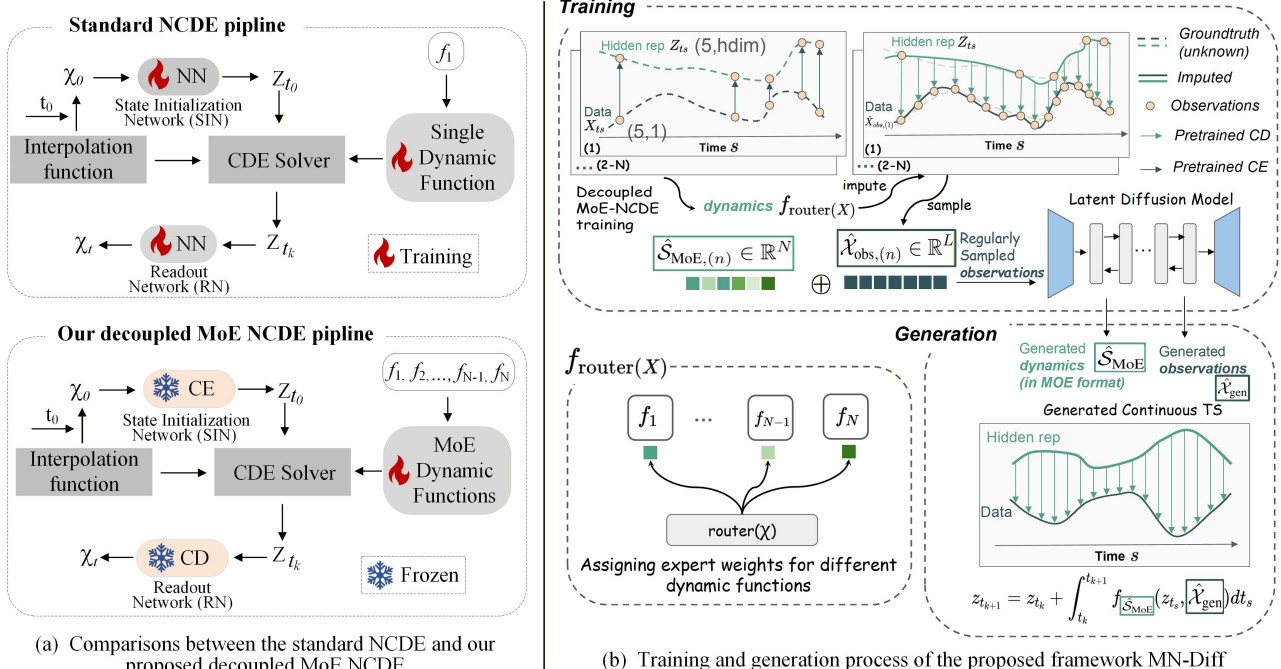

*Figure 2.* Overview of our method. Frozen CE and CD are pretrained channel-wise encoder and channel-wise decoder, respectively.

The *MoE dynamics* consist of a MLP (Multilayer Perceptron) based router $\mathcal{R} : \mathcal{O} \rightarrow \{0, 1, \ldots, N_e - 1\}$ and $N_e$ dynamic functions $f_\theta^{(i)}$ (each parameterized as a neural network), where $i \in [0, N_e - 1]$. The router $\mathcal{R}$ takes (potentially irregular) observations as input and outputs a routing score for each expert. Specifically, we first impute the irregular TS using cubic interpolation and then feed them into $\mathcal{R}$ to obtain all experts' scores for the $j$-th sample. The router is implemented as a simple MLP. A softmax function is then applied over the scores to produce a dense routing distribution. Given $\mathbf{r} \in \mathbb{R}^{N_e}$, the final output is computed as a weighted combination of all expert functions:

$$f_{\text{MoE}} = \sum_{i=0}^{N_e - 1} \mathbf{s}_i \cdot f_\theta^{(i)}, \quad \mathbf{s} = softmax(\mathbf{r}) \tag{4}$$

Finally, substituting $f_{\text{MoE}}$ into the standard NCDE formulas (Eq. 2 and Eq. 3) yields the MoE-NCDE.

Notably, the learned MoE weights explicitly encode the temporal dynamics of each training sample. This property provides a foundation for parameterizing the MoE via a diffusion model. Consequently, for newly generated samples, the diffusion model can produce adapted MoE weights that capture their temporal dynamics, enabling customized and effective continuous TSG. A detailed description will be provided in Section 3.4.

### 3.3. Decoupled Optimization Design for MoE-Dynamics-Focused Training

As shown in Figure 2(a), training a standard NCDE requires jointly optimizing the State Initialization Network (SIN), the Readout Network (RN), and the dynamic function network. Joint training can cause conflicts among modules and make the dynamic function network, which is the most critical component, difficult to learn effectively, especially when SIN and RN are poorly initialized.

To address this, we pretrain a channel-wise autoencoder on the irregular data and replace SIN and RN with the pretrained Channel-wise Encoder (CE) and Decoder (CD), keeping them frozen. This decoupled optimization (termed *Decoupled Design*) enables training to focus on the optimization of MoE dynamic functions while using CE and CD for a warm start, providing effective gradients and improving training stability. This is feasible because the pretrained CE and CD have the same input and output dimensions as SIN and RN, allowing direct substitution. The training of the CE and CD is shown in Appendix Algorithm 1 and overall MoE-dynamics-focused training in Appendix Algorithm 2.

### 3.4. Diffusion Parameterized MoE-NCDE for Continuous TSG

Building on the MoE-NCDE trained on a batch of samples $\mathcal{X} : [0, T]$, it can generate continuous time sequences by predicting values at any $\tilde{t} \in [0, T]$ consistent with contextual time information. However, the learned MoE weights,

which encode temporal dynamics, are specific to the training samples. New samples generated via a diffusion model have different temporal dynamics, rendering the original MoE weights incompatible. To address this, we propose parameterizing the MoE weights with a diffusion model, enabling newly generated samples to obtain adapted MoE weights that capture their temporal dynamics and allow customized, effective continuous TSG.

Specifically, the standard diffusion model $\mathcal{G}$ requires regular data, but our training samples $\mathcal{O}$ are irregular. We first use the pretrained MoE-NCDE to impute each sample into a regular sequence. The regularized samples, along with their corresponding MoE weights $s$, are then fed into the diffusion model to learn the joint distribution of temporal patterns and MoE weights. This parameterizes the MoE weights, allowing new samples $\tilde{\mathcal{O}}$ to receive adapted weights $\tilde{S}_{\text{MoE}}$ that describe their temporal dynamics. The training process is shown in the Appendix algorithm 3.

Finally, we input $\hat{S}_{\text{MoE}}$ into the pre-trained MoE-NCDE and perform continuous TSG for each sample using its generated MoE weights and `CDESolver`. The process is shown in Appendix Algorithm 4.

# 4. Experiments

The experiment aims to answer the following questions:

1. How effective is Diff-MN at irregular-to-regular TSG? (Section 4.2)
2. How to evaluate and how effective is Diff-MN at irregular-to-continuous TSG? (Section 4.3)
3. Are Diff-MN's tailored designs effective? (Section 4.4)
4. Does MoE decompose complex temporal dynamics for specialized learning and recombine them? (Table 5, Figure 6, and Appendix Table 14 discussed in Section 4.4)
5. Why is NCDE selected? (Appendix B.1 ) What is the impact of smooth interpolation functions? (Appendix B.2)

## 4.1. Experimental Settings

**Datasets.** The datasets include: (1) four public TSG datasets, Sines, Stocks, Energy, and MuJoCo; (2) ECG-related medical TS datasets from the UCR archive with diverse temporal patterns, including ECG200, ECG5K, ECGFD, and TLECG, with 2, 5,2 and 2 classes respectively; and (3) synthetic datasets, consisting of integrated signal waves and polynomial coefficient fitting data. For irregular modeling, we randomly discard 30%, 50%, and 70% of temporal observations to construct irregularly sampled datasets (Naiman et al., 2024). Details are provided in Appendix D.1.

**Baselines.** We compare our method with several representative advanced TSG methods, including irregular TSG

methods: KO-VAE (Naiman et al., 2024), GTGAN (Jeon et al., 2022), ProFITi (Yalavarthi et al., 2025) and HeT-VAE (Shukla & Marlin), as well as regular TSG methods: TimeGAN (Yoon et al., 2019), TimeVAE (Desai et al., 2021), and diffusion model (Huang et al., 2025). For regular TSG methods, we utilize standard NCDE to impute missing values before generation.

**Implementation details.** We conduct experiments using publicly available baseline code and train all models to convergence with recommended settings. In our method, the number of experts is fixed at four. More parameter settings, e.g., the diffusion model, are provided in the Appendix D.2.

For the four public datasets and two synthetic datasets, we construct training and evaluation samples with three sequence lengths (12, 24, and 36) and report averaged metrics. For the medical ECG classification datasets, samples are officially preprocessed, and default parameters are used.

**Evaluation metrics.** Following previous works (Naiman et al., 2024; Huang et al., 2025), we apply the following metrics for evaluating generation performance: (1) Discriminative score (DS), (2) Marginal distribution difference (MDD), (3) Kullback-Leibler divergence (KL), (4) Membership Inference Risk (MIR) metric (Tian et al., 2024) for privacy protection evaluation.

## 4.2. Performance of Irregular-to-Regular TSG

**Quantitative evaluation.** We first evaluate our model on the irregular TSG task. As shown in Tables 1 and 2, our method consistently outperforms baselines on both popular TSG and medical datasets across various missing rate settings. Although baselines handle irregular data via NCDE, their performance is limited by weak dynamic modeling and coupled optimization design. In contrast, our MoE-NCDE leverages a mixture-of-experts function and a dynamics-focused optimization strategy with a pre-trained channel-wise autoencoder, enabling more effective learning of temporal dynamics for irregular TS.

**Qualitative evaluation.** We also qualitatively evaluate the similarity between real and generated sequences using two visualizations: (i) t-SNE projections and (ii) kernel density estimation of probability density functions (PDFs). As shown in Figure 3, the t-SNE plots (top) exhibit strong overlap between real and synthetic samples, while the PDFs (bottom) show closer alignment with real data compared to KO-VAE and GT-GAN.

**Privacy protection analysis.** This experiment evaluates whether synthetic data generated by different models leaks information from real data, with results reported in Appendix Table 10. Our method not only achieves superior fidelity in DS, KL, and MDD but also exhibits the lowest privacy leakage risk according to MIR. This is due to the

*Table 1.* Irregular to regular TSG on popular datasets with 30%, 50%, and 70% of observations dropped. Performance is averaged over sequence lengths 12, 24, and 36. Results for each length are shown in Appendix Tables 7, 8, and 9.

| | | 30% | | | | 50% | | | | 70% | | | |
|---|---|---|---|---|---|---|---|---|---|---|---|---|---|
| | | Sines | Stocks | Energy | MuJoCo | Sines | Stocks | Energy | MuJoCo | Sines | Stocks | Energy | MuJoCo |
| **DS** | Ours | **0.105** | **0.142** | **0.422** | **0.293** | **0.128** | **0.137** | **0.487** | **0.375** | **0.182** | **0.106** | **0.497** | **0.393** |
| | KoVAE | 0.142 | 0.225 | 0.476 | 0.372 | 0.171 | 0.187 | 0.500 | 0.397 | 0.228 | 0.213 | 0.500 | 0.425 |
| | GT-GAN | 0.302 | 0.321 | 0.499 | 0.489 | 0.416 | 0.326 | 0.500 | 0.494 | 0.341 | 0.292 | 0.499 | 0.492 |
| | TimeGAN-NCDE | 0.455 | 0.365 | 0.499 | 0.49 | 0.427 | 0.442 | 0.499 | 0.499 | 0.44 | 0.477 | 0.500 | 0.499 |
| | TimeVAE-NCDE | 0.234 | 0.470 | 0.498 | 0.375 | 0.308 | 0.485 | 0.496 | 0.463 | 0.426 | 0.492 | 0.498 | 0.476 |
| | Diffusion-NCDE | 0.200 | 0.441 | 0.460 | 0.355 | 0.330 | 0.475 | 0.489 | 0.427 | 0.421 | 0.491 | 0.497 | 0.473 |
| | ProFITi | 0.307 | 0.454 | 0.495 | 0.492 | 0.344 | 0.483 | 0.499 | 0.497 | 0.392 | 0.489 | 0.5 | 0.497 |
| | HeTVAE | 0.491 | 0.328 | 0.497 | 0.499 | 0.5 | 0.443 | 0.5 | 0.499 | 0.5 | 0.487 | 0.5 | 0.5 |
| **MDD** | Ours | **0.953** | **0.25** | 0.270 | 0.347 | **1.093** | **0.281** | **0.252** | **0.318** | **1.308** | **0.299** | **0.279** | **0.297** |
| | KoVAE | 5.134 | 0.709 | 0.347 | 0.374 | 5.584 | 0.662 | 0.368 | 0.419 | 6.393 | 0.624 | 0.377 | 0.619 |
| | GT-GAN | 2.644 | 0.630 | 0.567 | 0.558 | 2.978 | 0.589 | 0.473 | 0.596 | 2.559 | 0.603 | 0.532 | 0.627 |
| | TimeGAN-NCDE | 3.477 | 1.111 | 1.028 | 1.182 | 3.739 | 1.605 | 1.343 | 1.535 | 3.981 | 1.629 | 1.559 | 1.704 |
| | TimeVAE-NCDE | 4.767 | 0.597 | 0.444 | 0.338 | 3.761 | 0.726 | 0.533 | 0.443 | 3.397 | 0.843 | 0.604 | 0.565 |
| | Diffusion-NCDE | 1.462 | 0.341 | 0.292 | 0.371 | 1.498 | 0.476 | 0.351 | 0.37 | 1.997 | 0.743 | 0.513 | 0.487 |
| | ProFITi | 1.163 | 0.297 | **0.255** | **0.295** | 1.236 | 0.408 | 0.35 | 0.353 | 1.443 | 0.445 | 0.442 | 0.448 |
| | HeTVAE | 3.158 | 0.599 | 0.625 | 0.654 | 3.449 | 0.773 | 1.095 | 1.093 | 3.967 | 1.032 | 1.323 | 1.398 |
| **KL** | Ours | **0.013** | **0.074** | **0.020** | 0.021 | **0.023** | **0.094** | **0.022** | **0.009** | **0.033** | **0.091** | **0.017** | **0.014** |
| | KoVAE | 4.172 | 1.014 | 0.065 | 0.130 | 4.719 | 0.859 | 0.067 | 0.198 | 5.521 | 0.868 | 0.08 | 0.425 |
| | GT-GAN | 0.099 | 0.203 | 0.056 | 0.021 | 0.153 | 0.181 | 0.051 | 0.028 | 0.163 | 0.223 | 0.057 | 0.048 |
| | TimeGAN-NCDE | 4.963 | 1.742 | 0.259 | 0.560 | 2.954 | 6.776 | 0.585 | 1.153 | 12.478 | 11.63 | 0.995 | 1.810 |
| | TimeVAE-NCDE | 1.999 | 0.224 | 0.088 | 0.075 | 1.487 | 0.259 | 0.101 | 0.118 | 1.196 | 0.345 | 0.136 | 0.172 |
| | Diffusion-NCDE | 0.056 | 0.085 | 0.065 | 0.045 | 0.070 | 0.094 | 0.058 | 0.058 | 0.095 | 0.105 | 0.087 | 0.078 |
| | ProFITi | 0.079 | 0.074 | 0.022 | **0.015** | 0.112 | 0.121 | 0.033 | 0.015 | 0.138 | 0.132 | 0.045 | 0.031 |
| | HeTVAE | 1.178 | 1.085 | 0.248 | 0.258 | 4.236 | 1.4 | 0.917 | 0.996 | 2.488 | 1.612 | 1.578 | 2.202 |

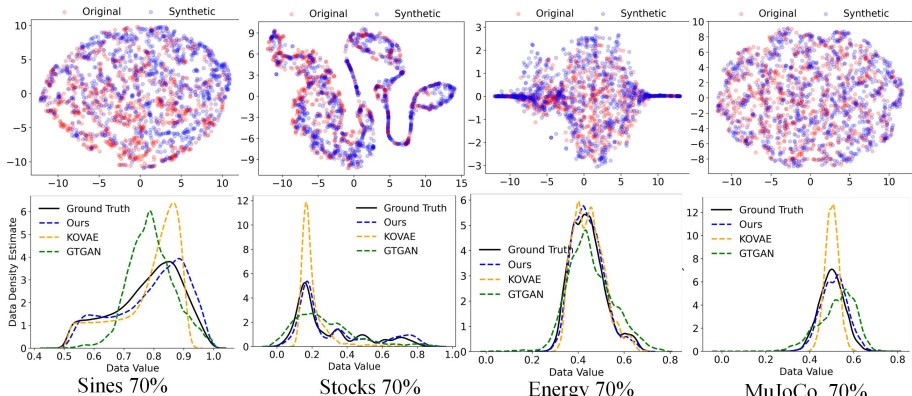

*Figure 3.* Qualitative evaluation with 2D t-SNE (top) and probability density function (PDF) of real, KoVAE, and GT-GAN (bottom) with 70% missing values. Additional cases in Appendix Figure 9.

MoE design, which promotes diverse generated samples rather than mere replication of the original data.

### 4.3. Performance of Irregular-to-Continuous TSG

**Refined continuous TSG on the downstream task.** In this experiment, we evaluate continuous TSG with irregular observations using a multivariate prediction task, which is conducted on the newly generated samples of each method. Baselines predict the last time step from length $t - 1$ data. For Diff-MN, we apply MoE-NCDE for newly generated samples and MoE weights from the diffusion model. For KOVAE and GTGAN, we apply the trained standard NCDE to their newly generated samples for Continuous TSG. Continuous TSG generates new points between each time step, creating a *refined sample* that roughly doubles the length of the historical window. Following previous downstream evaluations (Naiman et al., 2024; Jeon et al., 2022), the re-

fined sample is then used to train the Gated Recurrent Unit (GRU) to predict the same last step. We set $t = 12, 24, 36$ and report average MSE and MAE.

Table 3 shows that KOVAE-NCDE and GTGAN-NCDE perform worse than their non-continuous counterparts, indicating that standard NCDE for continuous generation is ineffective. In contrast, Ours-$\hat{S}_{\text{MoE}}$ outperforms its non-continuous version, improving forecasting performance. This is due to our diffusion parameterized MoE-NCDE, which generates MoE weights capturing the temporal dynamics of the newly generated samples, enabling better continuous generation. Visualizations of continuous TS are in Appendix Figures 10 to 12.

**Refined continuous TSG on solving for the analytical solutions of polynomial functions.** We further assess continuous TSG using a cubic function $y = ax^3 + bx^2 + cx + d$.

*Table 2.* Irregular to regular TSG on medical datasets (more diverse TS patterns) with 30%, 50%, and 70% of observations dropped.

| | | 30% | | | | 50% | | | | 70% | | | |
|---|---|---|---|---|---|---|---|---|---|---|---|---|---|
| | | ECG200 | ECG5K | ECGFD | TLECG | ECG200 | ECG5K | ECGFD | TLECG | ECG200 | ECG5K | ECGFD | TLECG |
| DS | Ours | **0.188** | **0.182** | **0.120** | **0.240** | **0.250** | **0.326** | **0.070** | **0.160** | **0.390** | **0.424** | **0.140** | **0.340** |
| | KoVAE | 0.500 | 0.500 | 0.420 | 0.400 | 0.485 | 0.498 | 0.410 | 0.440 | 0.485 | 0.500 | 0.380 | 0.400 |
| | GT-GAN | 0.467 | 0.493 | 0.450 | 0.490 | 0.457 | 0.495 | 0.380 | 0.470 | 0.475 | 0.497 | 0.430 | 0.490 |
| | TimeGAN-NCDE | 0.495 | 0.500 | 0.410 | 0.500 | 0.482 | 0.478 | 0.420 | 0.500 | 0.490 | 0.500 | 0.420 | 0.490 |
| | TimeVAE-NCDE | 0.440 | 0.371 | 0.350 | 0.430 | 0.427 | 0.370 | 0.380 | 0.440 | 0.415 | 0.400 | 0.400 | 0.450 |
| | Diffusion-NCDE | 0.458 | 0.334 | 0.390 | 0.430 | 0.460 | 0.396 | 0.430 | 0.430 | 0.458 | 0.429 | 0.380 | 0.400 |
| | ProFITi | 0.48 | 0.496 | 0.3 | 0.45 | 0.462 | 0.497 | 0.38 | 0.45 | 0.48 | 0.498 | 0.36 | 0.46 |
| | HeTVAE | 0.5 | 0.496 | 0.42 | 0.5 | 0.5 | 0.498 | 0.43 | 0.5 | 0.5 | 0.5 | 0.43 | 0.5 |
| MDD | Ours | **0.25** | **0.09** | **0.85** | **1.016** | **0.247** | **0.109** | **0.794** | **0.972** | **0.253** | **0.108** | **0.833** | **1.064** |
| | KoVAE | 0.551 | 0.364 | 1.588 | 1.467 | 0.565 | 0.359 | 1.581 | 1.439 | 0.523 | 0.357 | 1.604 | 1.475 |
| | GT-GAN | 0.495 | 0.409 | 0.97 | 1.158 | 0.506 | 0.437 | 0.968 | 1.091 | 0.556 | 0.296 | 0.99 | 1.177 |
| | TimeGAN-NCDE | 0.807 | 0.592 | 1.609 | 1.656 | 0.804 | 0.573 | 1.683 | 1.679 | 0.806 | 0.558 | 1.608 | 1.681 |
| | TimeVAE-NCDE | 0.417 | 0.173 | 1.026 | 1.136 | 0.37 | 0.175 | 0.970 | 1.265 | 0.363 | 0.154 | 0.939 | 1.131 |
| | Diffusion-NCDE | 0.389 | 0.187 | 0.945 | 1.125 | 0.369 | 0.190 | 0.973 | 1.082 | 0.328 | 0.198 | 0.958 | 1.130 |
| | ProFITi | 0.31 | 0.185 | 0.989 | 1.074 | 0.298 | 0.196 | 1.006 | 1.115 | 0.325 | 0.247 | 1.003 | 1.138 |
| | HeTVAE | 0.807 | 0.492 | 1.51 | 1.608 | 0.806 | 0.461 | 1.538 | 1.576 | 0.81 | 0.569 | 1.532 | 1.594 |
| KL | Ours | **0.053** | **0.014** | **0.052** | **0.129** | **0.09** | **0.021** | **0.099** | **0.073** | **0.113** | **0.018** | **0.04** | **0.153** |
| | KoVAE | 8.645 | 7.584 | 10.158 | 9.482 | 8.693 | 7.537 | 10.112 | 9.440 | 8.692 | 7.537 | 10.144 | 9.477 |
| | GT-GAN | 5.129 | 3.305 | 2.004 | 5.314 | 5.013 | 3.592 | 2.543 | 7.912 | 7.126 | 2.672 | 3.754 | 12.539 |
| | TimeGAN-NCDE | 17.57 | 16.615 | 13.165 | 17.182 | 17.56 | 14.408 | 13.154 | 17.191 | 17.57 | 9.431 | 13.165 | 17.191 |
| | TimeVAE-NCDE | 0.917 | 0.113 | 2.236 | 2.397 | 0.641 | 0.096 | 2.106 | 3.005 | 0.731 | 0.073 | 2.103 | 1.386 |
| | Diffusion-NCDE | 0.639 | 0.095 | 1.867 | 0.788 | 0.87 | 0.125 | 1.156 | 0.632 | 0.365 | 0.136 | 1.117 | 1.336 |
| | ProFITi | 0.085 | 0.053 | 0.204 | 0.245 | 0.073 | 0.044 | 0.262 | 0.346 | 0.149 | 0.066 | 0.395 | 1.188 |
| | HeTVAE | 17.216 | 8.574 | 8.218 | 16.49 | 17.199 | 6.251 | 13.445 | 16.41 | 17.193 | 10.316 | 13.445 | 16.458 |

*Table 3.* Diff-MN for continuous TSG on irregular time series (30%, 50%, 70% missing values). The performance is averaged over sequence lengths 12, 24, and 36. Results of each length are in Appendix Tables 11–13 and visualizations are in Appendix Figures 10 to 15.

| | | 30% | | | | 50% | | | | 70% | | | |
|---|---|---|---|---|---|---|---|---|---|---|---|---|---|
| | | Sines | Stocks | Energy | MuJoCo | Sines | Stocks | Energy | MuJoCo | Sines | Stocks | Energy | MuJoCo |
| MSE | Ours-$\hat{S}_{\text{MoE}}$ | **0.026** | **0.002** | **0.013** | **0.029** | **0.026** | **0.002** | **0.013** | **0.029** | **0.028** | **0.002** | **0.013** | **0.029** |
| | Ours | 0.049 | 0.005 | 0.014 | 0.031 | 0.05 | 0.005 | 0.014 | 0.030 | 0.054 | 0.005 | 0.014 | 0.030 |
| | KOVAE-NCDE | 0.076 | 0.041 | 0.023 | 0.030 | 0.076 | 0.036 | 0.021 | 0.032 | 0.079 | 0.0300 | 0.020 | 0.033 |
| | KOVAE | 0.044 | 0.025 | 0.014 | 0.030 | 0.044 | 0.019 | 0.014 | 0.031 | 0.045 | 0.014 | 0.014 | 0.032 |
| | GTGAN-NCDE | 0.089 | 0.021 | 0.027 | 0.053 | 0.076 | 0.018 | 0.026 | 0.053 | 0.070 | 0.021 | 0.028 | 0.058 |
| | GTGAN | 0.044 | 0.014 | 0.019 | 0.042 | 0.051 | 0.008 | 0.017 | 0.047 | 0.046 | 0.011 | 0.019 | 0.047 |
| MAE | Ours-$\hat{S}_{\text{MoE}}$ | **0.125** | **0.025** | **0.075** | **0.129** | **0.126** | **0.026** | **0.073** | **0.131** | **0.128** | **0.025** | **0.075** | **0.131** |
| | Ours | 0.174 | 0.036 | 0.075 | 0.137 | 0.176 | 0.036 | 0.074 | 0.136 | 0.182 | 0.036 | 0.076 | 0.136 |
| | KOVAE-NCDE | 0.199 | 0.132 | 0.114 | 0.139 | 0.200 | 0.123 | 0.110 | 0.142 | 0.204 | 0.116 | 0.105 | 0.146 |
| | KOVAE | 0.170 | 0.094 | 0.080 | 0.139 | 0.172 | 0.076 | 0.080 | 0.140 | 0.175 | 0.061 | 0.080 | 0.144 |
| | GTGAN-NCDE | 0.216 | 0.106 | 0.116 | 0.183 | 0.209 | 0.100 | 0.122 | 0.186 | 0.199 | 0.108 | 0.123 | 0.193 |
| | GTGAN | 0.176 | 0.069 | 0.093 | 0.160 | 0.186 | 0.056 | 0.089 | 0.174 | 0.174 | 0.057 | 0.099 | 0.173 |

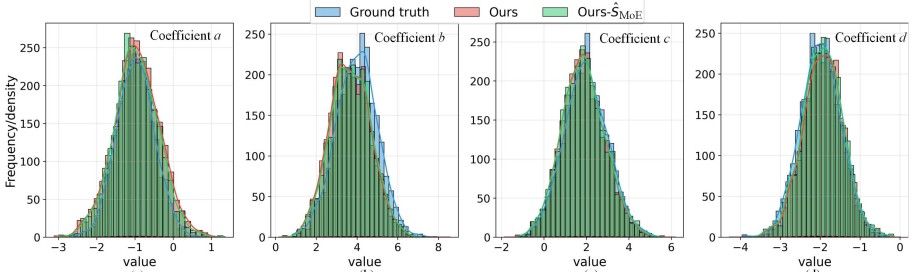

*Figure 4.* Cubic polynomial solutions of Diff-MN with 30% missing values: "Ours" uses regular generation from irregular data. "Ours-$\hat{S}_{\text{MoE}}$" applies generated MoE weights for continuous TSG (doubled length). Other cases are in Appendix Figures 16, 19, 20.

For each coefficient, 2500 samples are drawn from normal distributions with varying means and standard deviations. With $x$ linearly spaced over 24 values in $[-1, 1]$, $y$ forms time series (visualized in Appendix Figure 16). The model is trained on $y$, aiming for coefficients $\hat{a}, \hat{b}, \hat{c}, \hat{d}$ derived from generated $\hat{y}$ to match the original distributions.

Figure 4 shows that analytical solutions from our generated $\hat{y}$ closely match the true distribution. Both irregular-to-regular and irregular-to-continuous TSG preserve distributional similarity, with refined continuous generation

(Fig. 4(d)) closer to ground truth than non-continuous. This confirms the effectiveness of our method. In contrast, baselines using standard NCDE deviate significantly (Appendix Figures 17, 19, 20).

### 4.4. Ablation Study

**Quantitative evaluation.** Table 4 shows that removing MoE functions or the decoupled design significantly decreases model accuracy, and removing them together further deteriorates performance, indicating that these components

*Table 4.* Ablation study of MoE-NCDE on irregular medical ECG time series (30%, 50%, 70% missing).

|  |  | 30% | | | 50% | | | 70% | | |
|---|---|---|---|---|---|---|---|---|---|---|
|  |  | ECG200 | ECG5K | ECGFD | ECG200 | ECG5K | ECGFD | ECG200 | ECG5K | ECGFD |
| MDD | Ours | **0.211** | **0.062** | **0.424** | **0.237** | **0.079** | **0.507** | 0.246 | **0.095** | **0.624** |
|  | w/o MoE | 0.226 | 0.152 | 0.740 | 0.255 | 0.179 | 0.778 | **0.244** | 0.196 | 0.784 |
|  | w/o Decoupled Design | 0.370 | 0.191 | 0.878 | 0.397 | 0.174 | 0.850 | 0.361 | 0.154 | 0.865 |
|  | w/o MoE + w/o Decoupled Design | 0.412 | 0.173 | 0.965 | 0.363 | 0.175 | 0.968 | 0.361 | 0.157 | 0.984 |
| KL | Ours | **0.052** | **0.009** | **0.055** | **0.107** | **0.014** | **0.037** | 0.098 | **0.02** | **0.037** |
|  | w/o MoE | 0.097 | 0.091 | 0.068 | 0.111 | 0.13 | 0.037 | **0.082** | 0.155 | 0.076 |
|  | w/o Decoupled Design | 0.466 | 0.094 | 0.035 | 0.502 | 0.081 | 0.028 | 0.836 | 0.083 | 0.034 |
|  | w/o MoE + w/o Decoupled Design | 0.877 | 0.109 | 1.203 | 0.583 | 0.096 | 1.211 | 0.667 | 0.074 | 1.323 |

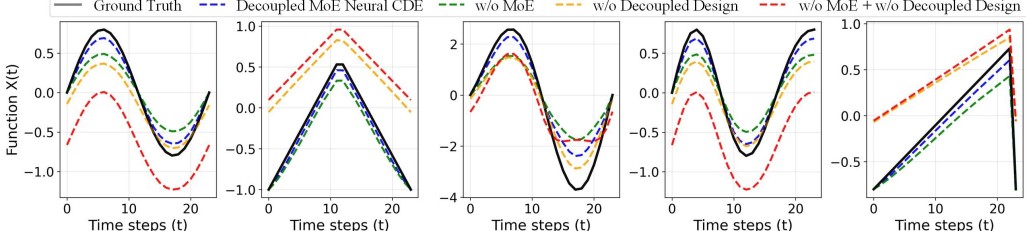

*Figure 5.* Qualitative results of MoE-NCDE ablation on various synthetic signals.

*Table 5.* Comparison of four-expert (8 linear layers) vs. single-expert (8 linear layers) on medical datasets.

|  |  | 30% | | | 50% | | | 70% | | |
|---|---|---|---|---|---|---|---|---|---|---|
|  |  | ECG200 | ECG5K | ECGFD | ECG200 | ECG5K | ECGFD | ECG200 | ECG5K | ECGFD |
| DS | Four-experts-eight-layers | **0.2** | **0.205** | **0.09** | **0.405** | **0.315** | **0.12** | **0.34** | **0.288** | **0.13** |
|  | One-expert-eight-layers | 0.328 | 0.364 | 0.13 | 0.44 | 0.389 | 0.23 | 0.392 | 0.412 | 0.21 |
| MDD | Four-experts-eight-layers | **0.211** | **0.062** | **0.424** | **0.237** | **0.079** | **0.529** | **0.246** | **0.095** | **0.613** |
|  | One-expert-eight-layers | 0.288 | 0.12 | 0.694 | 0.321 | 0.152 | 0.831 | 0.314 | 0.165 | 0.837 |
| KL | Four-experts-eight-layers | **0.052** | **0.009** | **0.055** | **0.107** | **0.014** | **0.05** | **0.098** | **0.02** | **0.049** |
|  | One-expert-eight-layers | 0.136 | 0.056 | 0.065 | 0.236 | 0.098 | 0.189 | 0.224 | 0.123 | 0.165 |

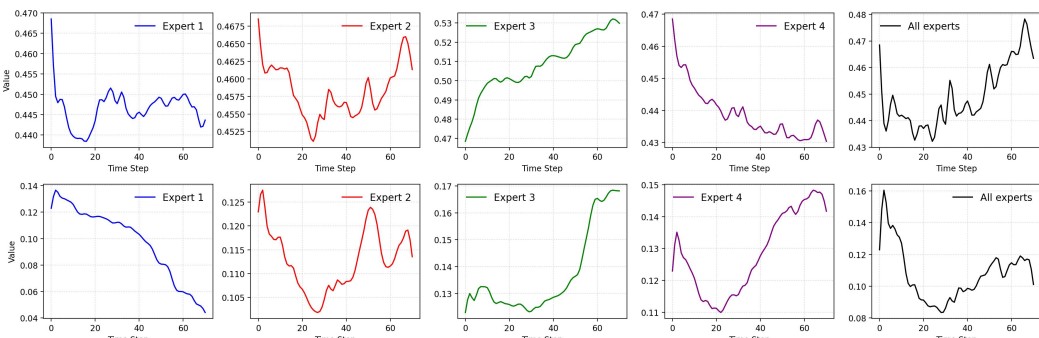

*Figure 6.* Visualization of temporal dynamics learned by different experts of MoE-NCDE on the **MuJoCo** dataset in continuous TSG. Energy dataset is shown in Appendix Figure 18.

are crucial. The MoE functions help NCDE learn temporal patterns, while the dynamics-focused decoupled optimization design (called *Decoupled Design*) uses a pre-trained channel-wise autoencoder to replace the network SIN and RN, allowing better focus on learning MoE dynamics.

**Qualitative evaluation.** We construct synthetic signals (e.g., sine, piecewise, sawtooth) and split them into training and test sets. We train MoE-NCDE and evaluate generation on the test set under component ablations (Fig. 5). Only with both MoE dynamics and the dynamics-focused decoupled optimization design does the model produce signals

resembling the real data; removing either causes high errors, confirming their effectiveness.

**Exploring why MoE is effective.** Table 5 compares NCDE performance between (i) four experts (each with one MLP and two linear layers, eight layers total) and (ii) a single expert with four stacked MLPs (same eight layers). The MoE approach significantly outperforms the single expert, as each expert specializes in learning distinct temporal dynamics, making the overall dynamics easier to learn while enhancing both expressiveness and generalization. Figure 6 further illustrates how MoE decomposes complex temporal

*Figure 7.* The impact of the number of experts (1-9) on the MDD metric. DS metric is shown in Appendix Figure 8.

patterns across experts and then integrates them, substantially enhancing NCDE's ability to model diverse temporal behaviors. Moreover, Appendix Table 14 reports the average expert weights across datasets. The weights are non-sparse, and different datasets favor different experts, indicating that the Mixture of Experts structure is effectively utilized rather than collapsing to a single expert.

**Dense vs. Sparse MoE Comparison.** To justify our choice of dense MoE, we conduct a systematic comparison between dense and sparse MoE under varying numbers of experts. For sparse MoE, we fix the total number of experts at 8 and activate 1, 2, 4, or 6 experts, respectively. All metrics are averaged across 30%, 50%, and 70% missing rates with sequence length 24.

*Table 6.* Expert collapse analysis on DS (↓) and MDD (↓). Metrics are averaged across 30%, 50%, 70% missing rates with sequence length 24. For sparse MoE, total experts = 8 with varying active experts. Numbers in parentheses indicate the number of experts used to achieve the best result.

| | DS (↓) | | | MDD (↓) | | |
|---|---|---|---|---|---|---|
| | ECG200 | ECG5K | ECGFD | ECG200 | ECG5K | ECGFD |
| Dense-1-Expert | 0.268 | 0.388 | **0.130** | 0.242 | 0.176 | 0.767 |
| Dense-2-Experts | **0.229** | 0.365 | 0.153 | **0.230** | 0.103 | 0.541 |
| Dense-4-Experts | 0.315 | **0.269** | 0.153 | 0.231 | 0.079 | **0.518** |
| Dense-6-Experts | 0.279 | 0.287 | 0.170 | 0.238 | **0.077** | 0.542 |
| Sparse-1-Expert | 0.335 | 0.292 | 0.177 | 0.235 | 0.085 | **0.542** |
| Sparse-2-Experts | 0.371 | **0.287** | **0.133** | 0.248 | 0.087 | 0.595 |
| Sparse-4-Experts | **0.288** | 0.291 | 0.173 | 0.231 | **0.082** | 0.568 |
| Sparse-6-Experts | **0.288** | 0.314 | 0.143 | **0.226** | 0.084 | 0.551 |
| Dense Best | **0.229** (2) | **0.269** (4) | **0.130** (1) | 0.230 (2) | **0.077** (6) | **0.518** (4) |
| Sparse Best | 0.288 (4) | 0.287 (2) | 0.133 (2) | **0.226** (6) | 0.082 (4) | 0.542 (1) |

As shown in Table 6, dense MoE achieves better overall performance (winning 7 vs. 5 on DS, 6 vs. 5 on MDD), where Win count is computed by pairwise Dense-vs-Sparse comparisons under the same expert count across all datasets. Dense MoE benefits from activating all experts, which provides more stable gradients and avoids routing to collapsed experts. Sparse MoE, which trains many experts but activates few during inference, is more susceptible to expert collapse under high data missingness, as insufficient data prevents effective specialization of individual experts.

### 4.5. Hyperparameter and Time Complexity Analysis

**Hyperparameter analysis.** Figure 7 shows that MoE-NCDE performance is not highly sensitive to the number of experts: using more than one improves performance, with

four experts yielding satisfactory results.

**Time complexity analysis.** MoE-NCDE has higher time complexity than standard NCDE (e.g., 1.2s vs. 0.81s for double-length refined generation on 3,674 stock samples of length 12) due to mixture-of-experts dynamics. However, by parameterizing the MoE weights via the diffusion model, we can directly generate sample-specific MoE weights for new data, avoiding NCDE retraining, additional feature extraction, and forward propagation, thereby reducing computational overhead.

## 5. Limitations

**MoE training under data missingness.** Both dense and sparse MoE degrade as expert count increases (Table 6), due to insufficient data and limited class diversity for specialization. Effective sparse MoE training and diversity-promoting mechanisms under high missingness remain open challenges.

**Scalability and data dependency.** The primary computational bottleneck lies in native NCDE rather than MoE overhead. Generation quality also depends on observation quality and the interpolation functions used to construct continuous input paths.

**Evaluation on naturally irregular data.** Following prior works, we simulate missingness and use held-out values as ground truth. Developing evaluation protocols for naturally irregular data where ground truth is unavailable remains future work.

## 6. Conclusion

This paper presents the first study of continuous TSG. Built upon NCDEs, Diff-MN introduces a decoupled MoE-NCDE that enhances standard NCDEs with mixture-of-experts dynamic functions and dynamics-focused optimization design, enabling more effective modeling of complex temporal dynamics and irregular TS for continuous generation. Moreover, we parameterize MoE weights via a diffusion model to generate sample-specific weights that reflect their temporal dynamics. Combined with a pre-trained NCDE, this enables adaptive continuous TSG. Diff-MN achieves state-of-the-art performance on continuous generation across multiple public and synthetic datasets.

## Impact Statement

This paper advances continuous time series generation applicable to diverse domains. While synthetic data generation offers benefits such as data augmentation and privacy-preserving sharing, it carries risks. For example, in healthcare, generated signals may contain hallucinated features that could mislead analysis if used without expert validation. Additionally, generative models may inadvertently memorize sensitive patterns from training data. We recommend applying appropriate privacy safeguards (e.g., differential privacy) before deploying on sensitive data, and caution that generated data should complement rather than replace real data in safety-critical applications.

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

# A. Supplementary Related Work

TSG for regular TS has gained increasing attention as a way to produce synthetic data for training, privacy preservation, and simulation. Early work explored adversarial learning approaches such as TimeGAN (Yoon et al., 2019), which leverage adversarial objectives to jointly capture temporal dynamics and sample realism. Variational methods have also been developed, including TimeVAE (Desai et al., 2021) and Vector Quantized TimeVAE (Lee et al., 2023). More recently, diffusion-based models (Lin et al., 2024) have emerged, synthesizing high-fidelity sequences by iteratively denoising random noise. These approaches illustrate the rapid methodological progress in generative modeling for TS. Recently, an increasing number of generation methods based on diffusion models have been proposed (Deng et al., 2025b; 2026; Huang et al., 2025; Xia et al., 2025; Li et al., 2025b), such as task-oriented generation, cross-domain generation, causal generation, and text-controlled generation. High-quality synthetic TS data is also critical for advancing downstream tasks such as long-term forecasting (Zhang et al., 2026b), predictability-aware training (Zhang et al., 2026a), and fine-tuning large pre-trained time series models (Zhang et al.), where data scarcity or distribution shift often limits performance.

# B. Discussion about Why Using Neural Controlled Differential Equation (NCDE)

## B.1. Comparisons between NODE and NCDE

### B.1.1. THEORETICAL PERSPECTIVE

Neural ordinary differential equation (NODE) (Chen et al., 2018) provides a continuous-time modeling framework for irregularly-sampled time series. The latent dynamics are governed by the following differential equation:

$$z(t) = z(0) + \int_0^t f_\theta(z(s)) \, ds, \tag{5}$$

where $z(s) \in \mathbb{R}^d$ is the latent state at time $s$, and $f_\theta : \mathbb{R}^d \to \mathbb{R}^d$ is a neural network parameterized by $\theta$ represent the dynamics of the differential equations.

However, NODE evolves the latent state solely from $z(0)$ based on learned dynamics, without direct conditioning on the observed values. Consequently, the latent trajectory may drift away from the true data distribution.

We adopt NCDE because Diff-MN requires both imputing irregular observations and performing continuous generation. Values generated between existing observation points need to be consistent with the temporal dynamics of the surrounding context; otherwise, the generated values may deviate from the true distribution of the underlying dynamical system.

### B.1.2. EXPERIMENTAL PERSPECTIVE

GT-GAN incorporates ODE-RNN and CTFP (Deng et al., 2020), where the latter borrows concepts from stochastic differential equation (SDE) and employs a Wiener process to introduce stochasticity. Ablation studies of GT-GAN consistently show that GT-GAN outperforms the two ODE-based methods. Additionally, KoVAE (a TSG method using NCDE) is reported to perform even better than GT-GAN, and our Diff-MN outperforms both GT-GAN and KoVAE. **All of this indicates that NCDE-based TSG under observed-data constraints outperforms ODE-based methods**.

## B.2. Proving interpolation control in NCDE Does Not Cause Over-smoothing

### B.2.1. THEORETICAL PERSPECTIVE

As discussed in the main text, NCDE can be defined as

$$z(t) = z(0) + \int_0^t f_\theta(z(s)) \, dX(s), \quad t \in [0, T], \tag{6}$$

where $X(s)$ is the control path (obtained via interpolation and smoothing), and $f_\theta$ is a nonlinear function (neural network).

The neural network $f_\theta$ can model highly nonlinear dynamics, amplifying or bending small variations in the smooth control path and thereby inducing complex, rapidly changing behavior in the latent trajectory.

Hence, even if $X(s)$ is smooth, the behavior of the integral result $z(t)$ depends on the properties of $f_\theta(z(s))$. If $f_\theta$ is highly nonlinear, the derivative $\frac{dz}{dt}$ can vary rapidly with $z(s)$, leading to non-smooth trajectories.

Specifically, when $X(t)$ is smooth and differentiable, the NCDE admits the following differential form:

$$\frac{dz(t)}{dt} = f_\theta(z(t)) \frac{dX(t)}{dt}. \tag{7}$$

Although the interpolated control path $X(t)$ is smooth by construction, the evolution of $z(t)$ is governed by the nonlinear function $f_\theta(z(t))$. As a result, even when $\frac{dX(t)}{dt}$ is smooth, the product $f_\theta(z(t))\frac{dX(t)}{dt}$ may vary rapidly due to the nonlinearity of $f_\theta$, leading to non-smooth latent trajectories.

In our model, the $f_\theta(z)$ is further parameterized as a Mixture-of-Experts:

$$f_\theta(z) = \sum_{i=1}^{N_e} \mathcal{R}_i(z) \, f_\theta^{(i)}(z), \tag{8}$$

where $\mathcal{R}_i(z) \geq 0$ and $\sum_{i=1}^{N_e} \mathcal{R}_i(z) = 1$ are routing weights produced by the router $\mathcal{R}$, and $f_\theta^{(i)}$ denotes the $i$-th expert.

Substituting the MoE formulation into the NCDE dynamics yields

$$\frac{dz(t)}{dt} = \left( \sum_{i=1}^{N_e} \mathcal{R}_i(z(t)) \, f_\theta^{(i)}(z(t)) \right) \frac{dX(t)}{dt}. \tag{9}$$

Since both the expert functions $f_\theta^{(i)}$ and the routing weights $\mathcal{R}_i(z(t))$ depend nonlinearly on the latent state, different experts may dominate the dynamics at different time instances. This allows the latent trajectory to exhibit rapid local variations and highly diverse temporal behaviors.

Therefore, even with a smooth interpolated control path $X(t)$, the NCDE equipped with MoE dynamics can generate expressive and potentially non-smooth latent trajectories.

### B.2.2. EXPERIMENTAL PERSPECTIVE

To validate this, we added visualizations of sequences generated solely by NCDE, as shown Figure 13, 14 and 15. These results show: **(i)** For intrinsically smooth data (e.g., sine in Figure 16), NCDE produces smooth trajectories and **(ii)** For intrinsically smooth data (e.g., sine in Figure 16), NCDE produces smooth trajectories.

Overall, while the control path is smooth, the nonlinear MoE–NCDE dynamics allow the generated sequences to exhibit fluctuating temporal behaviors, depending entirely on the underlying temporal dynamics.

## C. Supplementary Algorithm Pseudocodes

This presents the algorithm pseudocodes omitted from the Method section of the main text due to space constraints. As all algorithms have already been properly referenced and discussed, we simply include them here for convenient reference.

### C.1. Decoupled Optimization for MoE-Dynamics-Focused Training (Algorithm 2)

### C.2. Training Channel-wise Autoencoder (Algorithm 1)

### C.3. Joint Diffusion Training for $(\mathcal{O}_{\text{reg}}, s)$ to Parameterize MoE Weights (Algorithm 3)

### C.4. Continuous Time Series Generation (Algorithm 4)

## D. Supplementary Experimental Settings

### D.1. Datasets

We evaluate on four popular TSG datasets with diverse characteristics, following established benchmarks for generative time series (Yoon et al., 2019; Jeon et al., 2022). Sines is a simulated multivariate dataset where each channel follows a sinusoidal function with random frequency and phase, capturing continuous and periodic patterns. Stocks contains daily Google stock data (2004–2019) across six financial indicators, exhibiting aperiodic, random-walk behaviors. Energy is the

---

**Algorithm 1** Training channel-wise autoencoder

---

**Require:** Observed dataset $\mathcal{D}$, Control path $X$ (Cubic Spline Interpolation Function), *isnan* function and $E$ training epochs
1: Initialize parameterized channel-wise MLP based encoder $f_{CE}$ and decoder $f_{CD}$
2: **for** $\mathcal{O}^{(i)}$ in $\mathcal{D}$, repeat $E$ epochs **do**
3:     Get $\mathcal{O}^{(i)}_{\text{filled}}$ using $X$ to fill missing values in $\mathcal{O}^{(i)}$
4:     Get $\widetilde{\mathcal{O}}^{(i)}_{\text{filled}}$ by flattening $\mathcal{O}_{\text{filled}}$ from $[B, T, D]$ to $[B \times T, D]$ for channel-wise reconstruction
5:     **Encode and decode at channel-wise:** $h = f_{CE}(\mathcal{O}^{i}_{\text{filled}})$, $\mathcal{O}^{i}_{\text{recon}} = f_{CD}(h)$
6:     **Compute Loss** using non-missing values.
      $\mathcal{L} = \ell_{\text{MSE}}(\mathcal{O}^{i}_{\text{recon}}[\neg\text{mask}], \mathcal{O}^{i}[\neg\text{mask}])$.
7: **end for**

---

**Algorithm 2** Decoupled optimization for MoE-dynamics-focused training

---

**Require:** Observed dataset $\mathcal{D}$, an CDE solver `CDESolver`, $E$ training epochs.
1: Initialize $N$ parameterized dynamics experts $f_{\text{MoE}}$.
2: Initialize an MoE router $\mathcal{R}$.
3: Pretrained channel-wise encoder $f_{CE}$ and decoder $f_{CD}$ with forzen weights.
4: Control path $X$ (Cubic Spline Interpolation Function in this work).
5: **for** $\mathcal{O}^{(j)}$ in $\mathcal{D}$, repeat $E$ epochs **do**
6:     Get all expert weights **s** for $j$-th sample from $\mathcal{R}(\mathcal{O}^{(j)})$ using Softmax gating.
7:     $\hat{\mathcal{O}}^{(j)} \leftarrow$ `CDESolver(`
8:         $f_{\mathbf{s}}$, $f_{CE}(X(t_0^{(j)}))$, $\{t_0^{(j)}, t_1^{(j)}, \dots, t_{n_j}^{(j)}\}$, $X)$
9:     Gradient backward on $\ell_{MSE}(f_{CD}(\hat{\mathcal{O}}^{(j)}), \mathcal{O}^{(j)})$ to update $f_{\text{MoE}}$ (includes router $\mathcal{R}$) only.
10: **end for**

---

**Algorithm 3** Joint diffusion training for $(\mathcal{O}_{\text{reg}}, s)$ to parameterize MoE weights

---

**Require:** Dataset $\mathcal{D}_{\text{reg}}$, hyperparameter diffusion time step $T_d$ and the learned MoE weights $s$ for each sample
1: **while** not converged **do**
2:     Sample $x_0 = (\mathcal{O}_{\text{reg}}, s)$ from dataset $\mathcal{D}_{\text{reg}}$
3:     Randomly sample time step $n \sim \mathcal{U}(1, T_d)$
4:     Randomly sample noise $\epsilon \sim \mathcal{N}(0, I)$
5:     Corrupt data $x_t = \sqrt{\bar{\alpha}_t} \cdot x_0 + \sqrt{1 - \bar{\alpha}_t} \cdot \epsilon$
6:     Predict noise: $\hat{\epsilon} = \epsilon_\theta(x_t, t)$
7:     Compute loss: $\mathcal{L} \leftarrow \|\epsilon - \hat{\epsilon}\|^2$
8:     Update $\theta$
9: **end while**

---

**Algorithm 4** Continuous time series generation

---

**Require:** Observed dataset $\mathcal{D}$, and an CDE solver `CDESolver`
1: Train a $f_{\text{MoE}}$ with Algorithm 2
2: Impute the observed dataset $\mathcal{D}$ to $\mathcal{D}_{\text{reg}}$ with $f_{\text{MoE}}$
3: Train a diffusion model $\mathcal{G}$ with Algorithm 3
4: Generate $\hat{s}_i, \hat{\mathcal{O}}$ using diffusion model $\mathcal{G}$
5: $\hat{\mathcal{O}}^{(i)} =$ `CDESolver`$(f_{\hat{s}_i}, f_{CE}(\hat{X}(t_0^{(i)})), \{t_0^{(i)}, t_1^{(i)}, \dots, t_{n_i}^{(i)}\}, \hat{X})$
6: **return** $f_{CD}(\mathcal{O}^{(i)})$

---

UCI appliance energy prediction dataset (Candanedo, 2017), comprising 28 correlated channels with noisy periodicity and continuous-valued measurements. Finally, MuJoCo (**Mu**lti-**Jo**int dynamics with **Co**ntact) (Todorov et al., 2012) provides simulated physical dynamics with 14 channels.

The used medical ECG datasets are ECG200, ECG5000 (ECG5K), ECGFiveDays (ECGFD), and TwoLeadECG (TLECG),

which can be downloaded from the link[1]. Their sequence lengths are 96, 140, 136, and 82. The number of classes is 2, 5, 2, and 2, respectively. The data sizes are 200, 5000, 884, and 1162, respectively.

### D.2. More Implementation Details

When training MoE-NCDE, we use the Adam optimizer with a learning rate of $1e{-}3$, a batch size of 256, and a fixed hidden dimension of 64. For the TSG backbone of the standard diffusion model, the denoising network adopts a U-Net with four down- and up-sampling stages. Each stage contains two residual blocks and one cross-attention block, where residual blocks use two 1D convolution layers, and cross-attention employs 1D convolutions for input/output projections with eight attention heads. A middle block, placed between the down- and up-sampling paths, consists of two residual blocks and one attention block. In addition, we include input and output layers, each implemented with a single 1D convolution. SiLU is used as the activation function throughout. We train the model for 600 epochs with a batch size of 256, a learning rate of $1 \times 10^{-4}$.

## E. Supplementary Experimental Results

In Table 1 of the main text, we report the average performance of irregular-to-regular generation tasks across sequence lengths of 12, 24, and 36. Here, we additionally present results for each individual length. Overall, our method consistently outperforms others, in line with the conclusions from Table 1, and we therefore omit further discussion. We simply include them here for convenient reference.

**E.1. Irregular to Regular TSG on Popular Datasets with Sequence Length 12 (Table 7)**

**E.2. Irregular to Regular TSG on Popular Datasets with Sequence Length 24 (Table 8)**

**E.3. Irregular to Regular TSG on Popular Datasets with Sequence Length 36 (Table 9)**

**E.4. TSNE and PDF Visualizations on Irregular to Regular TSG (Figure 9)**

**E.5. Evaluation of Privacy Protection Capability (Table 10)**

**E.6. Visualizations about MoE-NCDE Not Producing Overly smooth curves (Figure 13, 14 and 15)**

**E.7. Irregular to Continuous TSG on Popular Datasets with Sequence Length 12 (Table 11)**

**E.8. Irregular to Continuous TSG on Popular Datasets with Sequence Length 24 (Table 12)**

**E.9. Irregular to Continuous TSG on Popular Datasets with Sequence Length 36 (Table 13)**

**E.10. Visualization of Continuous TSG on the Stock Dataset (Figure 10 for Part 1, Figure 11 for Part 2, and Figure 12 for Part 3)**

**E.11. Visualization of Cubic Polynomial Curves (Figure 16)**

**E.12. Cubic Polynomial Solutions of Diff-MN with 50% and 70% Missing Values (Figure 17)**

**E.13. Cubic Polynomial Solutions of Diff-MN VS. KOVAE with 30%, 50% and 70% Missing Values (Figure 19)**

**E.14. Cubic Polynomial Solutions of Diff-MN VS. GT-GAN with 30%, 50% and 70% Missing Values (Figure 20)**

**E.15. Visualization of Temporal Dynamics Learned by Different Experts (Figure 18)**

**E.16. Average Expert Weights Learned by MoE-NCDE (Figure 18)**

**E.17. Impact of the Number of Experts on the DS Metric (Figure 8)**

---

[1]www.timeseriesclassification.com

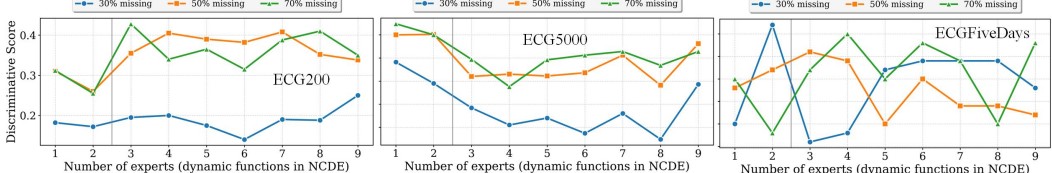

*Figure 8.* The impact of the number of experts (1-9) on the DS metric.

*Table 7.* Irregular to regular TSG on popular datasets with **sequence length 12**.

|  |  | 30% | | | | 50% | | | | 70% | | | |
|---|---|---|---|---|---|---|---|---|---|---|---|---|---|
|  |  | Sines | Stocks | Energy | MuJoCo | Sines | Stocks | Energy | MuJoCo | Sines | Stocks | Energy | MuJoCo |
| DS | Ours | **0.077** | **0.084** | **0.386** | **0.268** | **0.13** | **0.042** | **0.472** | **0.417** | **0.222** | **0.054** | **0.494** | **0.45** |
|  | KoVAE | 0.218 | 0.103 | 0.500 | 0.358 | 0.202 | 0.106 | 0.500 | 0.390 | 0.211 | 0.228 | 0.500 | 0.451 |
|  | GT-GAN | 0.300 | 0.385 | 0.499 | 0.498 | 0.423 | 0.357 | 0.499 | 0.496 | 0.434 | 0.370 | 0.498 | 0.497 |
|  | TimeGAN-NCDE | 0.468 | 0.369 | 0.498 | 0.487 | 0.422 | 0.472 | 0.499 | 0.499 | 0.429 | 0.479 | 0.500 | 0.498 |
|  | TimeVAE-NCDE | 0.354 | 0.493 | 0.496 | 0.433 | 0.466 | 0.497 | 0.491 | 0.477 | 0.500 | 0.497 | 0.498 | 0.480 |
|  | Diffusion-NCDE | 0.264 | 0.428 | 0.426 | 0.323 | 0.353 | 0.472 | 0.482 | 0.437 | 0.480 | 0.491 | 0.497 | 0.475 |
|  | ProFITi | 0.29 | 0.44 | 0.492 | 0.49 | 0.339 | 0.478 | 0.498 | 0.496 | 0.41 | 0.486 | 0.5 | 0.497 |
|  | HeTVAE | 0.5 | 0.368 | 0.493 | 0.499 | 0.5 | 0.478 | 0.499 | 0.5 | 0.5 | 0.493 | 0.5 | 0.5 |
| MDD | Ours | **1.015** | **0.16** | **0.303** | **0.303** | **1.337** | **0.2** | **0.251** | **0.319** | **1.783** | **0.219** | **0.319** | **0.34** |
|  | KoVAE | 10.137 | 0.289 | 0.343 | 0.326 | 10.12 | 0.288 | 0.377 | 0.372 | 10.13 | 0.312 | 0.399 | 0.635 |
|  | GT-GAN | 3.574 | 0.746 | 0.598 | 0.540 | 3.898 | 0.600 | 0.414 | 0.565 | 3.38 | 0.733 | 0.562 | 0.585 |
|  | TimeGAN-NCDE | 4.599 | 1.109 | 1.020 | 1.261 | 4.799 | 1.625 | 1.395 | 1.485 | 5.35 | 1.717 | 1.499 | 1.761 |
|  | TimeVAE-NCDE | 8.661 | 0.712 | 0.451 | 0.463 | 6.045 | 0.864 | 0.587 | 0.586 | 5.184 | 1.054 | 0.652 | 0.707 |
|  | Diffusion-NCDE | 2.150 | 0.328 | 0.306 | 0.276 | 2.128 | 0.538 | 0.365 | 0.428 | 2.902 | 0.854 | 0.506 | 0.508 |
|  | ProFITi | 1.277 | 0.32 | 0.254 | 0.298 | 1.342 | 0.416 | 0.342 | 0.361 | 1.584 | 0.443 | 0.442 | 0.446 |
|  | HeTVAE | 3.594 | 0.597 | 0.635 | 0.507 | 4.25 | 0.748 | 1.11 | 1.092 | 5.134 | 0.986 | 1.273 | 1.105 |
| KL | Ours | **0.007** | **0.05** | **0.019** | **0.017** | **0.014** | **0.075** | **0.025** | **0.018** | **0.033** | **0.066** | **0.013** | **0.024** |
|  | KoVAE | 9.941 | 0.139 | 0.076 | 0.168 | 9.511 | 0.149 | 0.084 | 0.282 | 9.788 | 0.151 | 0.111 | 0.579 |
|  | GT-GAN | 0.144 | 0.283 | 0.063 | 0.020 | 0.187 | 0.212 | 0.050 | 0.029 | 0.164 | 0.314 | 0.066 | 0.031 |
|  | TimeGAN-NCDE | 5.230 | 1.742 | 0.210 | 0.575 | 1.125 | 7.401 | 0.716 | 0.997 | 15.176 | 12.148 | 0.854 | 2.397 |
|  | TimeVAE-NCDE | 3.551 | 0.298 | 0.089 | 0.158 | 2.776 | 0.345 | 0.128 | 0.245 | 2.113 | 0.534 | 0.148 | 0.338 |
|  | Diffusion-NCDE | 0.004 | 0.012 | 0.004 | 0.007 | 0.028 | 0.040 | 0.019 | 0.036 | 0.074 | 0.095 | 0.046 | 0.042 |
|  | ProFITi | 0.078 | 0.084 | 0.022 | 0.016 | 0.082 | 0.124 | 0.032 | 0.016 | 0.11 | 0.137 | 0.044 | 0.032 |
|  | HeTVAE | 0.036 | 0.302 | 0.376 | 0.102 | 2.655 | 0.46 | 1.124 | 0.934 | 0.288 | 0.78 | 1.275 | 1.079 |

*Table 8.* Irregular to regular TSG on popular datasets with **sequence length 24**.

|  |  | 30% | | | | 50% | | | | 70% | | | |
|---|---|---|---|---|---|---|---|---|---|---|---|---|---|
|  |  | Sines | Stocks | Energy | MuJoCo | Sines | Stocks | Energy | MuJoCo | Sines | Stocks | Energy | MuJoCo |
| DS | Ours | **0.069** | **0.171** | **0.414** | **0.249** | **0.072** | **0.151** | **0.491** | **0.306** | **0.127** | **0.179** | **0.499** | **0.347** |
|  | KoVAE | 0.055 | 0.285 | 0.500 | 0.358 | 0.090 | 0.208 | 0.500 | 0.363 | 0.212 | 0.236 | 0.500 | 0.403 |
|  | GT-GAN | 0.276 | 0.305 | 0.500 | 0.480 | 0.338 | 0.325 | 0.500 | 0.491 | 0.286 | 0.273 | 0.500 | 0.489 |
|  | TimeGAN-NCDE | 0.457 | 0.351 | 0.500 | 0.486 | 0.433 | 0.454 | 0.500 | 0.497 | 0.438 | 0.472 | 0.499 | 0.499 |
|  | TimeVAE-NCDE | 0.191 | 0.481 | 0.499 | 0.324 | 0.175 | 0.493 | 0.499 | 0.462 | 0.431 | 0.490 | 0.499 | 0.464 |
|  | Diffusion-NCDE | 0.148 | 0.460 | 0.455 | 0.373 | 0.286 | 0.490 | 0.489 | 0.409 | 0.392 | 0.489 | 0.496 | 0.470 |
|  | ProFITi | 0.298 | 0.459 | 0.498 | 0.496 | 0.365 | 0.487 | 0.499 | 0.498 | 0.409 | 0.492 | 0.5 | 0.498 |
|  | HeTVAE | 0.48 | 0.306 | 0.498 | 0.499 | 0.5 | 0.401 | 0.5 | 0.499 | 0.5 | 0.476 | 0.5 | 0.5 |
| MDD | Ours | **0.752** | **0.269** | **0.218** | **0.336** | **0.775** | **0.251** | **0.216** | **0.341** | **0.971** | **0.299** | **0.265** | **0.285** |
|  | KoVAE | 2.756 | 0.905 | 0.347 | 0.341 | 3.314 | 0.833 | 0.361 | 0.380 | 5.221 | 0.763 | 0.368 | 0.633 |
|  | GT-GAN | 2.257 | 0.551 | 0.429 | 0.576 | 2.467 | 0.563 | 0.396 | 0.532 | 2.277 | 0.532 | 0.413 | 0.555 |
|  | TimeGAN-NCDE | 3.162 | 1.134 | 1.046 | 1.173 | 3.555 | 1.597 | 1.226 | 1.553 | 3.609 | 1.592 | 1.605 | 1.651 |
|  | TimeVAE-NCDE | 3.408 | 0.580 | 0.418 | 0.307 | 3.042 | 0.709 | 0.521 | 0.411 | 2.862 | 0.804 | 0.597 | 0.527 |
|  | Diffusion-NCDE | 1.303 | 0.401 | 0.250 | 0.336 | 1.402 | 0.403 | 0.356 | 0.338 | 1.667 | 0.692 | 0.557 | 0.486 |
|  | ProFITi | 0.961 | 0.311 | 0.261 | 0.284 | 1.446 | 0.472 | 0.358 | 0.333 | 1.645 | 0.436 | 0.444 | 0.407 |
|  | HeTVAE | 2.775 | 0.582 | 0.607 | 0.722 | 3.431 | 0.787 | 0.993 | 1.09 | 3.677 | 1.071 | 1.229 | 1.445 |
| KL | Ours | **0.011** | **0.084** | **0.009** | **0.007** | **0.022** | **0.092** | **0.014** | **0.006** | **0.035** | **0.105** | **0.023** | **0.013** |
|  | KoVAE | 1.696 | 1.347 | 0.058 | 0.150 | 1.861 | 1.205 | 0.061 | 0.198 | 2.840 | 1.380 | 0.071 | 0.508 |
|  | GT-GAN | 0.065 | 0.163 | 0.042 | 0.032 | 0.139 | 0.161 | 0.053 | 0.019 | 0.190 | 0.184 | 0.051 | 0.034 |
|  | TimeGAN-NCDE | 4.573 | 1.380 | 0.227 | 0.666 | 4.339 | 5.887 | 0.489 | 1.248 | 4.854 | 10.508 | 1.119 | 1.409 |
|  | TimeVAE-NCDE | 2.18 | 0.209 | 0.067 | 0.046 | 1.291 | 0.245 | 0.102 | 0.070 | 1.038 | 0.284 | 0.132 | 0.112 |
|  | Diffusion-NCDE | 0.060 | 0.438 | 0.073 | 0.033 | 0.053 | 0.440 | 0.099 | 0.041 | 0.077 | 0.498 | 0.114 | 0.047 |
|  | ProFITi | 0.128 | 0.091 | 0.028 | 0.01 | 0.137 | 0.161 | 0.036 | 0.012 | 0.189 | 0.123 | 0.043 | 0.026 |
|  | HeTVAE | 0.323 | 0.792 | 0.175 | 0.318 | 8.712 | 1.203 | 0.495 | 0.915 | 0.337 | 1.398 | 1.429 | 2.343 |

*Table 9.* Irregular to regular TSG on popular datasets with **sequence length 36**.

| | | 30% | | | | 50% | | | | 70% | | | |
|---|---|---|---|---|---|---|---|---|---|---|---|---|---|
| | | Sines | Stocks | Energy | MuJoCo | Sines | Stocks | Energy | MuJoCo | Sines | Stocks | Energy | MuJoCo |
| DS | Ours | **0.169** | **0.170** | **0.466** | **0.363** | **0.183** | **0.218** | **0.499** | **0.403** | **0.196** | **0.085** | **0.499** | **0.381** |
| | KoVAE | 0.152 | 0.286 | 0.429 | 0.399 | 0.222 | 0.247 | 0.500 | 0.439 | 0.261 | 0.176 | 0.500 | 0.422 |
| | GT-GAN | 0.331 | 0.272 | 0.499 | 0.489 | 0.487 | 0.297 | 0.500 | 0.494 | 0.303 | 0.234 | 0.500 | 0.489 |
| | TimeGAN-NCDE | 0.440 | 0.376 | 0.499 | 0.496 | 0.425 | 0.399 | 0.498 | 0.500 | 0.453 | 0.48 | 0.500 | 0.499 |
| | TimeVAE-NCDE | 0.157 | 0.437 | 0.500 | 0.368 | 0.282 | 0.464 | 0.499 | 0.451 | 0.347 | 0.490 | 0.498 | 0.485 |
| | Diffusion-NCDE | 0.189 | 0.435 | 0.498 | 0.368 | 0.352 | 0.462 | 0.496 | 0.436 | 0.391 | 0.493 | 0.497 | 0.474 |
| | ProFITi | 0.342 | 0.481 | 0.499 | 0.498 | 0.354 | 0.493 | 0.5 | 0.498 | 0.357 | 0.494 | 0.5 | 0.498 |
| | HeTVAE | 0.494 | 0.309 | 0.499 | 0.498 | 0.5 | 0.449 | 0.5 | 0.499 | 0.5 | 0.493 | 0.5 | 0.5 |
| MDD | Ours | **1.091** | **0.322** | **0.288** | **0.402** | **1.167** | **0.392** | **0.288** | **0.293** | **1.17** | **0.38** | **0.253** | **0.267** |
| | KoVAE | 2.509 | 0.933 | 0.350 | 0.456 | 3.319 | 0.865 | 0.366 | 0.506 | 3.828 | 0.798 | 0.363 | 0.588 |
| | GT-GAN | 2.101 | 0.593 | 0.673 | 0.558 | 2.568 | 0.604 | 0.610 | 0.692 | 2.021 | 0.543 | 0.620 | 0.74 |
| | TimeGAN-NCDE | 2.670 | 1.091 | 1.019 | 1.111 | 2.863 | 1.592 | 1.408 | 1.568 | 2.984 | 1.578 | 1.574 | 1.7 |
| | TimeVAE-NCDE | 2.232 | 0.499 | 0.463 | 0.243 | 2.195 | 0.604 | 0.491 | 0.331 | 2.144 | 0.670 | 0.564 | 0.462 |
| | Diffusion-NCDE | 0.934 | 0.295 | 0.319 | 0.308 | 0.963 | 0.487 | 0.331 | 0.345 | 1.421 | 0.682 | 0.475 | 0.468 |
| | ProFITi | 0.935 | 0.252 | 0.259 | 0.289 | 1.024 | 0.394 | 0.366 | 0.337 | 1.161 | 0.45 | 0.444 | 0.452 |
| | HeTVAE | 3.104 | 0.619 | 0.634 | 0.734 | 2.665 | 0.784 | 1.182 | 1.098 | 3.091 | 1.038 | 1.466 | 1.644 |
| KL | Ours | **0.020** | **0.089** | **0.031** | **0.039** | **0.033** | **0.115** | **0.026** | **0.003** | **0.031** | **0.101** | **0.014** | **0.004** |
| | KoVAE | 0.878 | 1.555 | 0.061 | 0.072 | 2.785 | 1.224 | 0.057 | 0.115 | 3.935 | 1.072 | 0.057 | 0.188 |
| | GT-GAN | 0.087 | 0.163 | 0.064 | 0.012 | 0.134 | 0.171 | 0.051 | 0.035 | 0.134 | 0.171 | 0.054 | 0.078 |
| | TimeGAN-NCDE | 5.086 | 2.104 | 0.341 | 0.440 | 3.398 | 7.039 | 0.551 | 1.214 | 17.403 | 12.235 | 1.012 | 1.625 |
| | TimeVAE-NCDE | 0.265 | 0.166 | 0.107 | 0.021 | 0.395 | 0.186 | 0.074 | 0.038 | 0.437 | 0.216 | 0.128 | 0.067 |
| | Diffusion-NCDE | 0.087 | 0.235 | 0.151 | 0.042 | 0.083 | 0.334 | 0.044 | 0.071 | 0.123 | 0.398 | 0.163 | 0.039 |
| | ProFITi | 0.08 | 0.054 | 0.022 | 0.012 | 0.172 | 0.115 | 0.036 | 0.011 | 0.192 | 0.121 | 0.047 | 0.029 |
| | HeTVAE | 3.174 | 2.16 | 0.194 | 0.353 | 1.34 | 2.537 | 1.131 | 1.14 | 6.838 | 2.659 | 2.03 | 3.184 |

*Table 10.* Privacy-leakage risk (MIR metric) of different methods on medical datasets (lower is better)

| | | 30% | | | | 50% | | | | 70% | | | | |
|---|---|---|---|---|---|---|---|---|---|---|---|---|---|---|
| | | ECG200 | ECG5K | ECGFD | TLECG | ECG200 | ECG5K | ECGFD | TLECG | ECG200 | ECG5K | ECGFD | TLECG | Avg. |
| MIR | Ours | **0.6826** | 0.6944 | **0.6734** | **0.6671** | **0.6671** | 0.6676 | 0.6866 | 0.7302 | 0.697 | **0.6866** | 0.6765 | 0.6866 | **0.6846** |
| | KoVAE | 0.6873 | **0.6849** | 0.7194 | 0.6882 | 0.6761 | 0.6873 | 0.8846 | 0.8679 | 0.7931 | 0.7077 | 0.697 | 0.697 | 0.7325 |
| | GT-GAN | 0.6993 | 0.8772 | 0.9709 | 0.9794 | 0.7698 | 0.7628 | 0.9583 | 0.9787 | **0.6765** | 0.7541 | 0.807 | 0.7667 | 0.8334 |
| | TimeGAN-NCDE | 0.7463 | 0.7463 | 0.7463 | 0.6817 | 0.7163 | 0.8482 | 0.807 | 0.807 | 0.807 | 0.7667 | 0.7419 | 0.807 | 0.7685 |
| | TimeVAE-NCDE | 0.7194 | 0.7067 | 0.6897 | 0.6676 | 0.6676 | 0.6698 | 0.7541 | 0.7541 | 0.7797 | 0.7419 | 0.7419 | 0.7419 | 0.7195 |
| | Diffusion-NCDE | 0.7067 | 0.7067 | 0.6873 | 0.6693 | 0.6698 | 0.6698 | 0.7541 | 0.7797 | 0.8364 | 0.7419 | 0.7188 | 0.7302 | 0.7226 |
| | ProFITi | 0.7067 | 0.7042 | 0.7194 | 0.6988 | 0.6798 | 0.6969 | 0.7797 | 0.902 | 0.807 | 0.7797 | 0.7188 | 0.8846 | 0.7565 |
| | HetVAE | 0.7692 | 0.6993 | 0.7117 | 0.7981 | 0.8163 | 0.8299 | 0.7931 | 0.8214 | 0.807 | 0.7188 | 0.7302 | 0.8519 | 0.7789 |

*Table 11.* Diff-MN for continuous TSG on irregular time series (30%, 50%, 70% missing) with sequence length 12.

| | | 30% | | | | 50% | | | | 70% | | | |
|---|---|---|---|---|---|---|---|---|---|---|---|---|---|
| | | Sines | Stocks | Energy | MuJoCo | Sines | Stocks | Energy | MuJoCo | Sines | Stocks | Energy | MuJoCo |
| MSE | Ours-$\hat{S}_{\text{MoE}}$ | **0.016** | **0.002** | **0.014** | **0.027** | **0.017** | **0.002** | **0.013** | **0.029** | **0.017** | **0.002** | **0.014** | **0.027** |
| | Ours | 0.030 | 0.004 | 0.014 | 0.025 | 0.031 | 0.005 | 0.014 | 0.027 | 0.034 | 0.004 | 0.014 | 0.026 |
| | KOVAE-NCDE | 0.070 | 0.013 | 0.03 | 0.028 | 0.070 | 0.012 | 0.028 | 0.029 | 0.07 | 0.011 | 0.027 | 0.031 |
| | KOVAE | 0.036 | 0.006 | 0.014 | 0.027 | 0.036 | 0.006 | 0.014 | 0.027 | 0.036 | 0.006 | 0.014 | 0.028 |
| | GTGAN-NCDE | 0.064 | 0.039 | 0.029 | 0.041 | 0.052 | 0.028 | 0.024 | 0.040 | 0.060 | 0.028 | 0.030 | 0.040 |
| | GTGAN | 0.034 | 0.029 | 0.018 | 0.032 | 0.043 | 0.011 | 0.016 | 0.031 | 0.033 | 0.026 | 0.018 | 0.030 |
| MAE | Ours-$\hat{S}_{\text{MoE}}$ | **0.107** | **0.025** | **0.077** | **0.124** | **0.109** | **0.025** | **0.074** | **0.127** | **0.111** | **0.024** | **0.076** | **0.126** |
| | Ours | 0.140 | 0.034 | 0.077 | 0.124 | 0.143 | 0.035 | 0.075 | 0.128 | 0.149 | 0.033 | 0.077 | 0.127 |
| | KOVAE-NCDE | 0.190 | 0.085 | 0.137 | 0.132 | 0.190 | 0.083 | 0.134 | 0.134 | 0.190 | 0.079 | 0.128 | 0.141 |
| | KOVAE | 0.157 | 0.039 | 0.080 | 0.131 | 0.157 | 0.040 | 0.080 | 0.130 | 0.156 | 0.041 | 0.080 | 0.135 |
| | GTGAN-NCDE | 0.193 | 0.147 | 0.111 | 0.162 | 0.181 | 0.122 | 0.115 | 0.162 | 0.183 | 0.122 | 0.121 | 0.16 |
| | GTGAN | 0.153 | 0.110 | 0.090 | 0.141 | 0.174 | 0.077 | 0.087 | 0.142 | 0.15 | 0.103 | 0.093 | 0.139 |

*Table 12.* Diff-MN for continuous TSG on irregular time series (30%, 50%, 70% missing) with sequence length 24.

| | | 30% | | | | 50% | | | | 70% | | | |
|---|---|---|---|---|---|---|---|---|---|---|---|---|---|
| | | Sines | Stocks | Energy | MuJoCo | Sines | Stocks | Energy | MuJoCo | Sines | Stocks | Energy | MuJoCo |
| MSE | Ours-$\hat{S}_{\text{MoE}}$ | **0.02** | **0.003** | **0.013** | **0.027** | **0.020** | **0.002** | **0.013** | **0.027** | **0.022** | **0.002** | **0.013** | **0.027** |
| | Ours | 0.055 | 0.007 | 0.013 | 0.030 | 0.055 | 0.006 | 0.013 | 0.029 | 0.059 | 0.006 | 0.014 | 0.029 |
| | KOVAE-NCDE | 0.080 | 0.053 | 0.021 | 0.030 | 0.080 | 0.049 | 0.018 | 0.032 | 0.084 | 0.039 | 0.018 | 0.033 |
| | KOVAE | 0.044 | 0.033 | 0.014 | 0.029 | 0.044 | 0.022 | 0.014 | 0.030 | 0.047 | 0.018 | 0.014 | 0.031 |
| | GTGAN-NCDE | 0.077 | 0.010 | 0.02 | 0.056 | 0.079 | 0.01 | 0.023 | 0.053 | 0.073 | 0.013 | 0.022 | 0.058 |
| | GTGAN | 0.047 | 0.007 | 0.017 | 0.038 | 0.047 | 0.005 | 0.016 | 0.048 | 0.048 | 0.003 | 0.016 | 0.047 |
| MAE | Ours-$\hat{S}_{\text{MoE}}$ | **0.106** | **0.025** | **0.074** | **0.131** | **0.107** | **0.025** | **0.072** | **0.129** | **0.109** | **0.025** | **0.074** | **0.131** |
| | Ours | 0.179 | 0.040 | 0.074 | 0.138 | 0.18 | 0.037 | 0.073 | 0.137 | 0.187 | 0.038 | 0.075 | 0.135 |
| | KOVAE-NCDE | 0.191 | 0.152 | 0.109 | 0.140 | 0.192 | 0.144 | 0.096 | 0.143 | 0.199 | 0.135 | 0.096 | 0.146 |
| | KOVAE | 0.166 | 0.119 | 0.08 | 0.137 | 0.167 | 0.079 | 0.08 | 0.139 | 0.176 | 0.071 | 0.080 | 0.143 |
| | GTGAN-NCDE | 0.19 | 0.079 | 0.104 | 0.188 | 0.199 | 0.078 | 0.115 | 0.188 | 0.197 | 0.086 | 0.108 | 0.194 |
| | GTGAN | 0.179 | 0.052 | 0.087 | 0.157 | 0.173 | 0.045 | 0.084 | 0.177 | 0.173 | 0.034 | 0.086 | 0.173 |

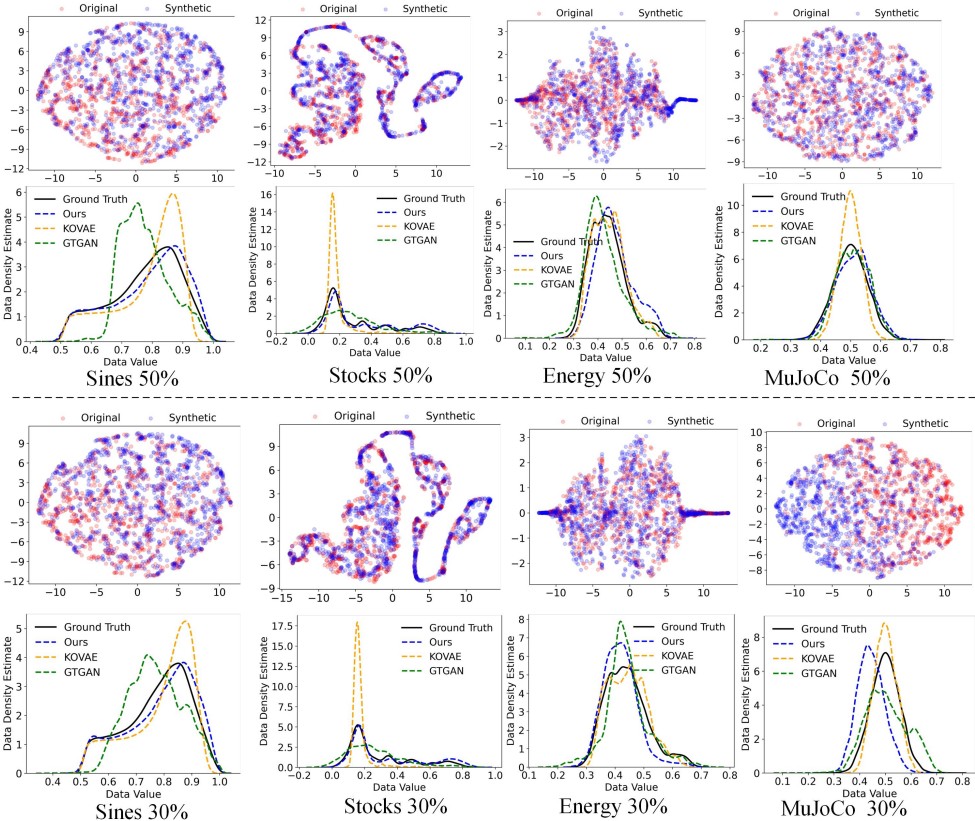

*Figure 9.* Qualitative evaluation with 2D t-SNE and probability density function (PDF) of real, KoVAE, and GT-GAN with 50% and 30% observations dropped.

*Table 13.* Diff-MN for continuous TSG on irregular time series (30%, 50%, 70% missing) with sequence length 36.

| | | | 30% | | | | 50% | | | | 70% | | |
|---|---|---|---|---|---|---|---|---|---|---|---|---|---|
| | | Sines | Stocks | Energy | MuJoCo | Sines | Stocks | Energy | MuJoCo | Sines | Stocks | Energy | MuJoCo |
| MSE | Ours-$\hat{S}_{\text{MoE}}$ | **0.041** | **0.002** | **0.013** | **0.031** | **0.042** | **0.002** | **0.013** | **0.031** | **0.044** | **0.003** | **0.013** | **0.032** |
| | Ours-NCDE | 0.041 | 0.002 | 0.013 | 0.031 | 0.042 | 0.002 | 0.013 | 0.031 | 0.044 | 0.003 | 0.013 | 0.033 |
| | Ours | 0.063 | 0.005 | 0.013 | 0.038 | 0.064 | 0.005 | 0.014 | 0.035 | 0.068 | 0.005 | 0.014 | 0.036 |
| | KOVAE-NCDE | 0.077 | 0.057 | 0.018 | 0.033 | 0.078 | 0.046 | 0.018 | 0.035 | 0.082 | 0.039 | 0.017 | 0.036 |
| | KOVAE | 0.051 | 0.035 | 0.014 | 0.035 | 0.052 | 0.029 | 0.014 | 0.036 | 0.051 | 0.018 | 0.014 | 0.038 |
| | GTGAN-NCDE | 0.126 | 0.013 | 0.032 | 0.061 | 0.098 | 0.015 | 0.031 | 0.065 | 0.076 | 0.021 | 0.034 | 0.076 |
| | GTGAN | 0.052 | 0.006 | 0.022 | 0.054 | 0.064 | 0.007 | 0.020 | 0.062 | 0.058 | 0.004 | 0.023 | 0.065 |
| MAE | Ours-$\hat{S}_{\text{MoE}}$ | **0.162** | **0.026** | **0.074** | **0.133** | **0.161** | **0.027** | **0.074** | **0.135** | **0.163** | **0.026** | **0.075** | **0.137** |
| | Ours | 0.203 | 0.035 | 0.074 | 0.149 | 0.204 | 0.036 | 0.074 | 0.144 | 0.209 | 0.036 | 0.076 | 0.145 |
| | KOVAE-NCDE | 0.214 | 0.159 | 0.096 | 0.146 | 0.218 | 0.141 | 0.099 | 0.149 | 0.225 | 0.133 | 0.091 | 0.151 |
| | KOVAE | 0.186 | 0.123 | 0.080 | 0.150 | 0.191 | 0.111 | 0.081 | 0.151 | 0.193 | 0.072 | 0.081 | 0.155 |
| | GTGAN-NCDE | 0.266 | 0.093 | 0.132 | 0.198 | 0.247 | 0.099 | 0.135 | 0.208 | 0.218 | 0.115 | 0.141 | 0.226 |
| | GTGAN | 0.197 | 0.046 | 0.103 | 0.182 | 0.209 | 0.048 | 0.098 | 0.202 | 0.201 | 0.034 | 0.117 | 0.208 |

*Table 14.* Average expert weights learned by MoE-NCDE across all samples from different datasets.

| Datasets | Expert 1 | Expert 2 | Expert 3 | Expert 4 |
|---|---|---|---|---|
| Sines | **0.31384414** | 0.2231202 | 0.26881662 | 0.19421977 |
| Stocks | 0.19200034 | 0.21767974 | **0.32304975** | 0.2672697 |
| Energy | 0.2413441 | **0.31066164** | 0.2065838 | 0.24141027 |
| MuJoCo | 0.18345672 | 0.241069 | 0.2046078 | **0.370866** |
| ECG200 | 0.21229412 | **0.44273788** | 0.20304906 | 0.14191894 |
| ECG5K | 0.21999635 | 0.182932 | **0.3610456** | 0.2360259 |
| ECGFD | 0.17836495 | **0.31938562** | 0.2739047 | 0.2283447 |
| TLECG | 0.19264506 | 0.2577268 | **0.3210036** | 0.2286246 |

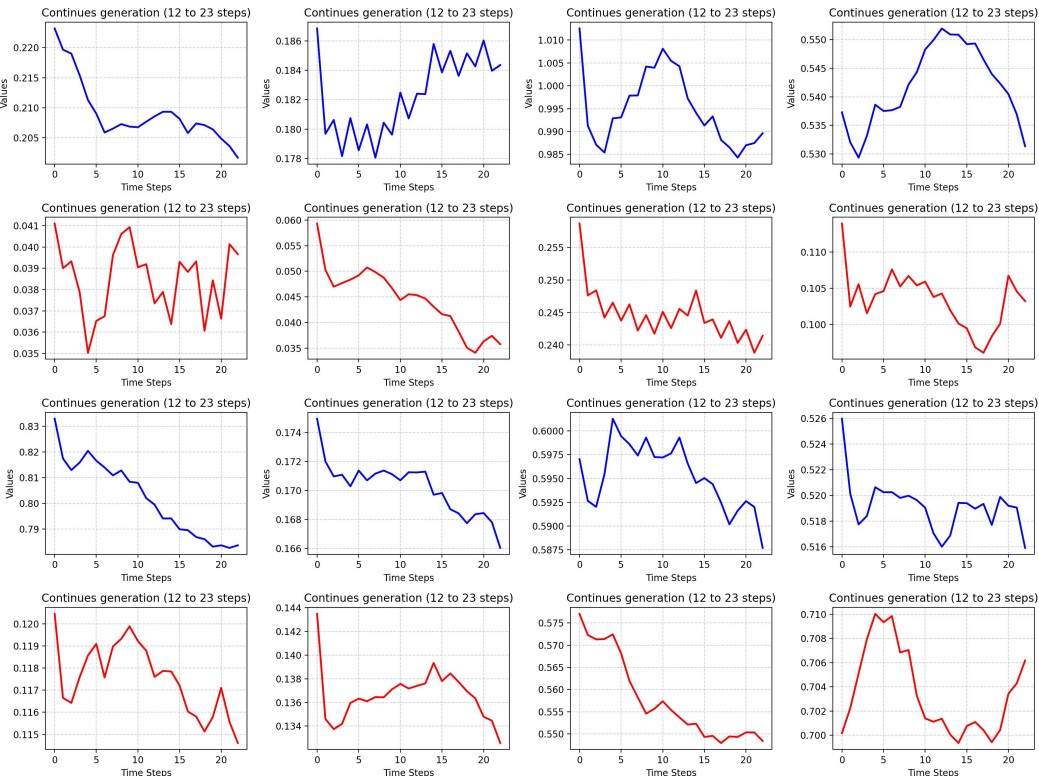

*Figure 10.* Continuous TSG on the **stock dataset** with Diff-MN: **12 steps refined to 23** under 30% missing data **(Part 1)**.

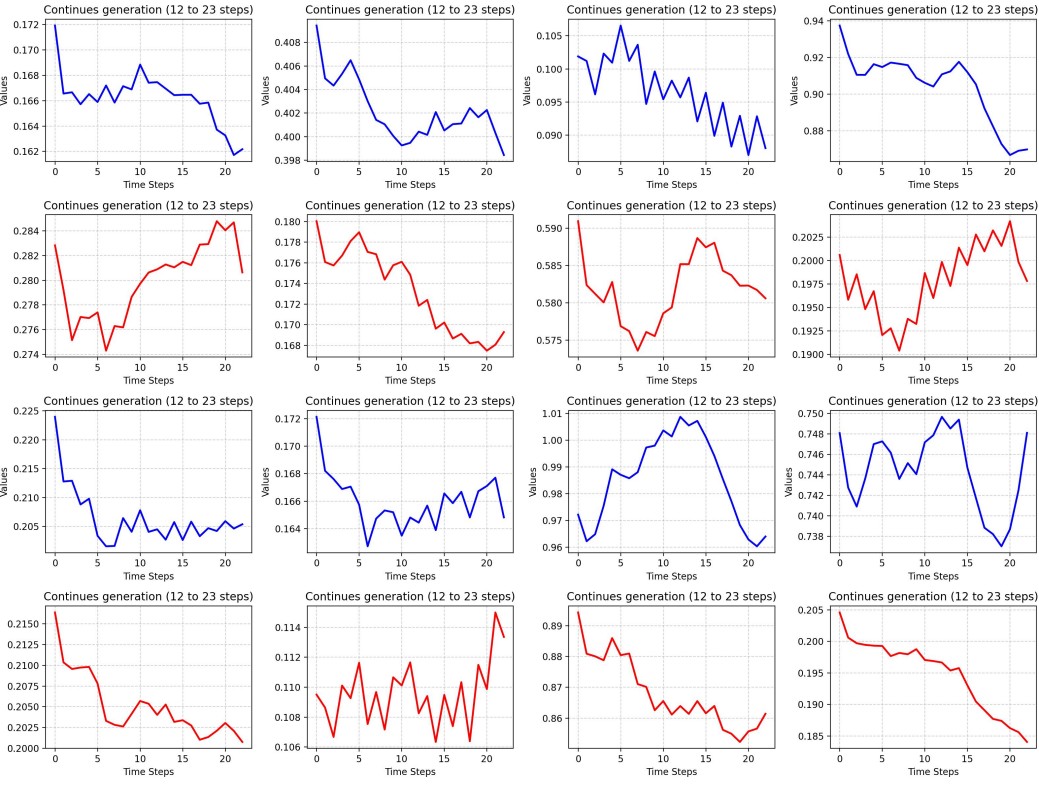

*Figure 11.* Continuous TSG on the **stock dataset** with Diff-MN: **12 steps refined to 23** under 30% missing data **(Part 2)**.

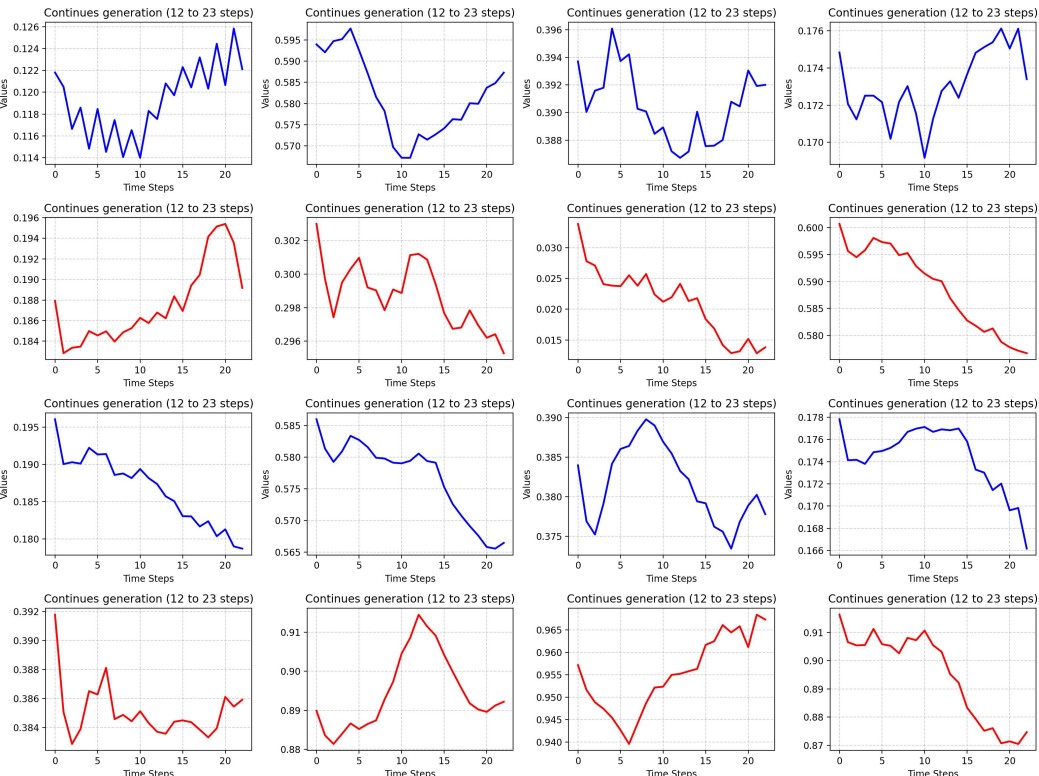

*Figure 12.* Continuous TSG on the stock dataset with Diff-MN: **12 steps refined to 23** under 30% missing data (**Part 3**).

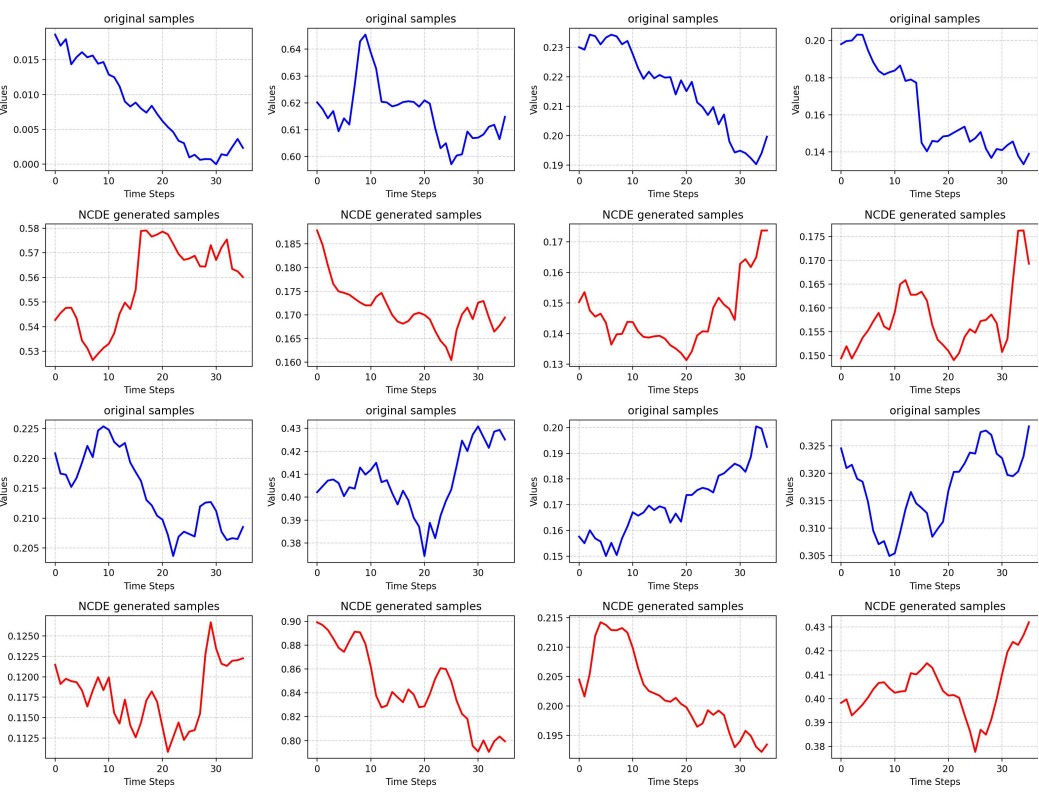

*Figure 13.* Visualizations demonstrate that MoE-NCDE **does not produce overly smooth curves** on the **stock dataset**.

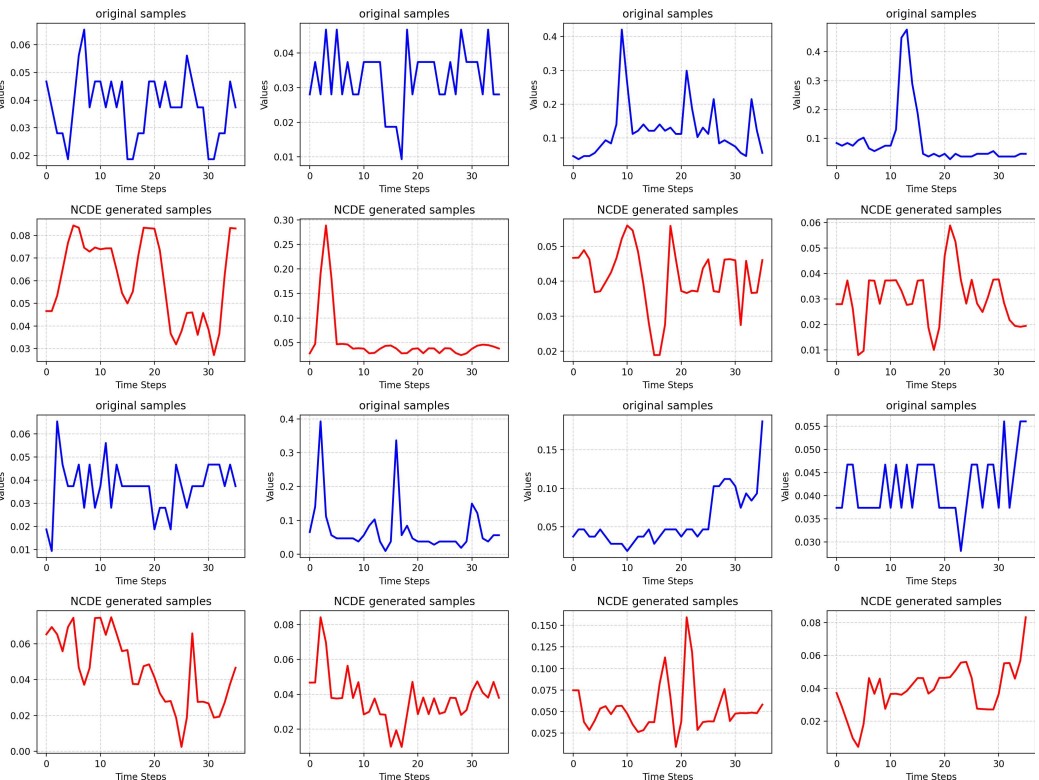

*Figure 14.* Visualizations demonstrate that MoE-NCDE **does not produce overly smooth curves** on the **energy dataset**.

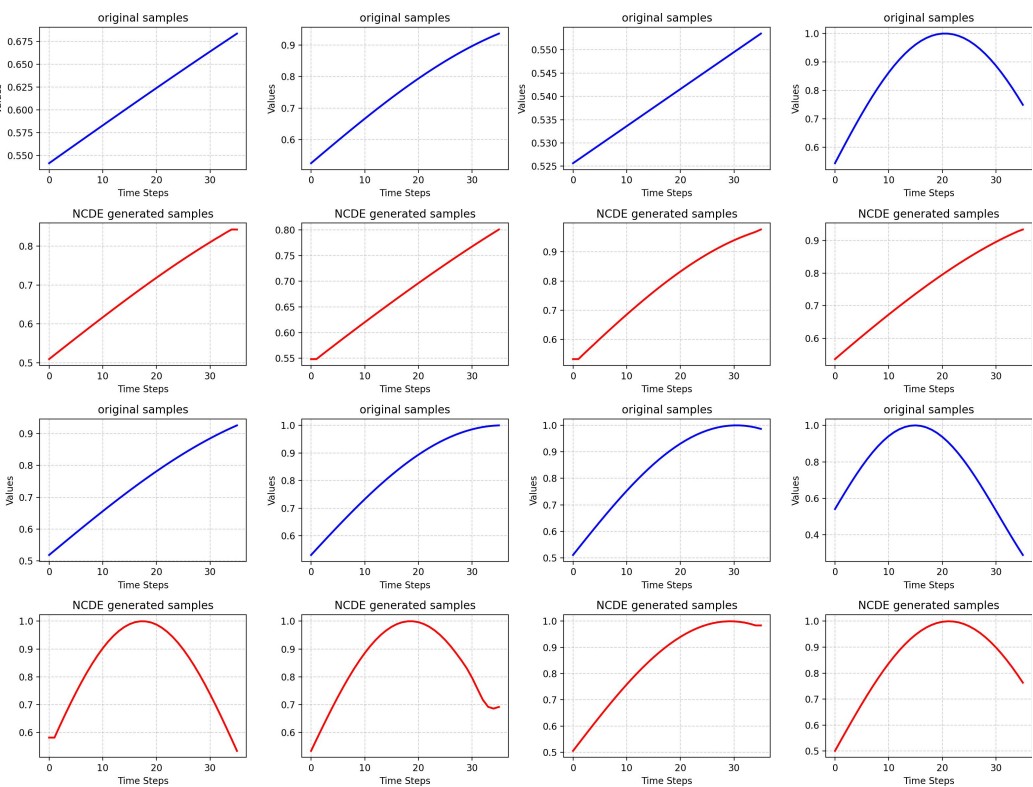

*Figure 15.* Visualizations demonstrate that MoE-NCDE **does not produce overly smooth curves** on the **sine dataset**.

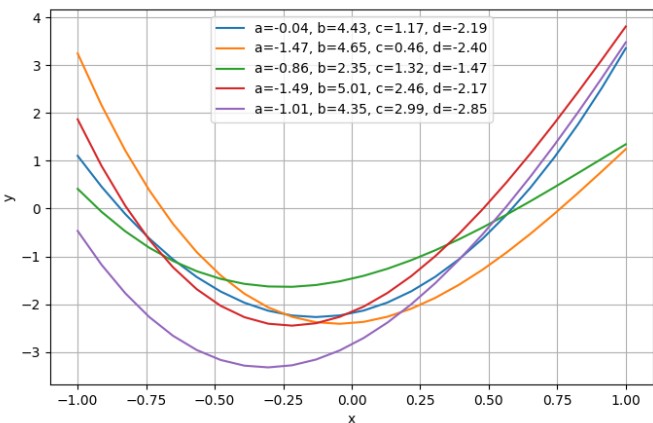

*Figure 16.* Visualization of cubic polynomial curves with varying coefficients.

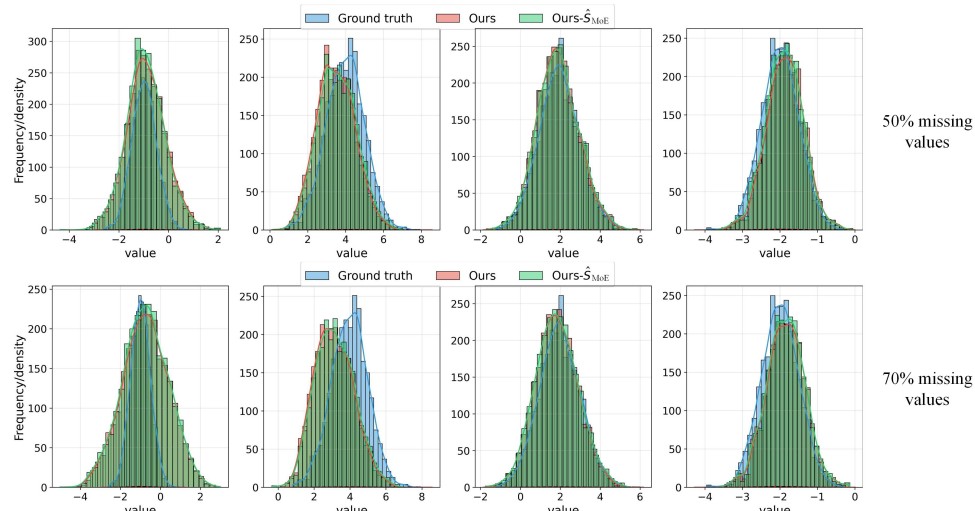

*Figure 17.* Analytical solution recovery using continuous TSG samples (doubled length). Subplots show coefficients $a$, $b$, $c$, and $d$.

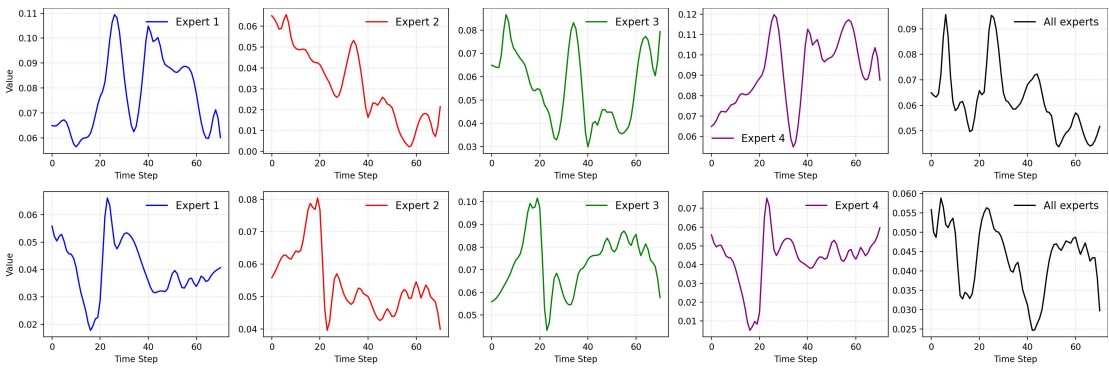

*Figure 18.* Visualization of temporal dynamics learned by different experts of MoE-NCDE on the **Energy** dataset in continuous TSG.

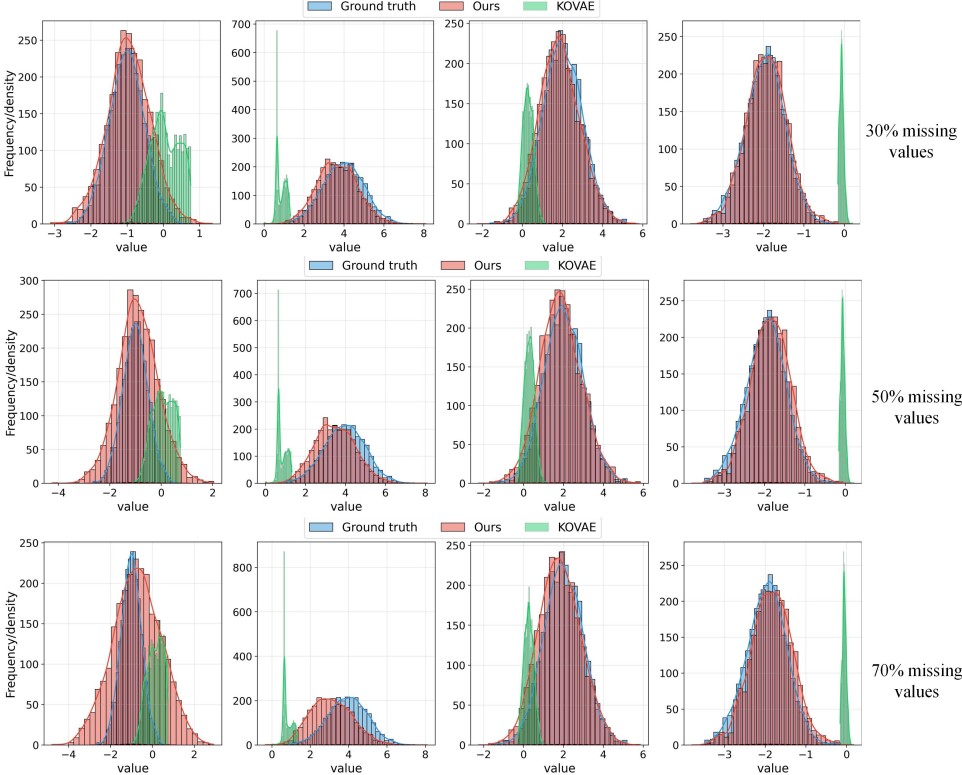

*Figure 19.* Comparison of analytical solution recovery between our method and KOVAE. Subplots show coefficients $a$, $b$, $c$, and $d$.

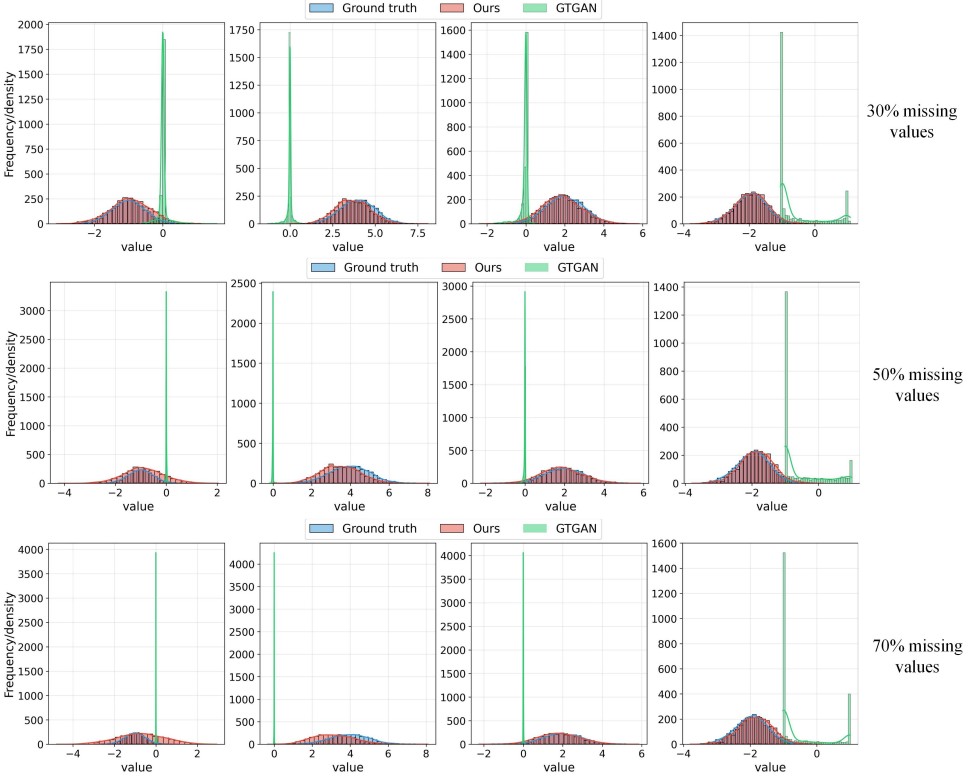

*Figure 20.* Comparison of analytical solution recovery between our method and GT-GAN. Subplots show coefficients $a$, $b$, $c$, and $d$.

