# OpenReview forum: "Diff-MN: Diffusion Parameterized  MoE-NCDE for Continuous Time Series Generation with Irregular Observations"
_ICML.cc/2026/Conference — ICML 2026 regular_

### Official Review · Reviewer_4pP9 · 2026-03-08

**Soundness:** 2
**Presentation:** 3
**Significance:** 2
**Originality:** 3
**Overall Recommendation:** 4
**Confidence:** 4

**Summary:**

The paper proposes MN-Diff, a framework for continuous time series generation from irregular observations. The approach combines a Mixture-of-Experts (MoE) architecture with Neural Controlled Differential Equations (NCDE) to model dynamic temporal dependencies and generate continuous-time trajectories. In addition, it employs a decoupled training strategy to enhance parameter optimization and improve training stability. Experimental results on several real-world and synthetic datasets indicate that the proposed method achieves improved performance compared with baseline models.

**Compliance With Llm Reviewing Policy:**

Affirmed.

**Final Justification:**

The authors' rebuttal has addressed my concerns, and I will raise my initial score.

**Key Questions For Authors:**

1. Why does the single-expert model with 8 linear layers in Table 5, which has a parameter count comparable to the standard MN-Diff, exhibit lower MDD performance than the single-expert model with only 2 linear layers shown in Figure 7?
2. How does the model scale on longer sequences (128, 256), which are more commonly encountered in real-world scenarios?

**Limitations:**

yes

**Strengths And Weaknesses:**

Strengths:
1. The work focuses on learning from irregularly time series and generating continuous-time trajectories, a scenario relatively unexplored in prior work. The problem is well-motivated and relevant to real-world applications such as finance and healthcare.
2. The experimental setup is fairly comprehensive, covering multiple types of datasets and evaluation metrics. It evaluates the generated samples from several perspectives, including quality, practical value, and privacy preservation, and reports improvements over baseline models. The implementation details facilitate the reproducibility of the proposed method.


Weaknesses:
1. While NCDE, MoE and decoupled training are widely used in prior works, the novelty of this paper lies in their combination to irregular time series rather than introducing new theoretical mechanisms. The contribution is thus incremental.
2. Conventional MoE architectures typically select the top-k experts sparsely, whereas MN-Diff employs a dense MoE. The rationale for this choice is not further experimentally validated.
3. As all experts are trained under the same framework without mechanisms to enforce diversity, they tend to capture overlapping and redundant patterns. This is corroborated by Figure 7, where changes in the number of experts do not lead to further performance gains.

---

> ### Author Rebuttal · Authors · 2026-03-31
>
> **Weaknesses**
>
> **W1: the novelty is the combination of existing components without introducing new theoretical mechanisms.**
>
> **A1:**
>
> Actually, our work goes beyond a simple integration of existing components, offering meaningful insights while addressing a challenging task:
>
> (1) Task novelty. We are the first to formally explore continuous TSG, including both generation techniques and evaluation protocols.
>
> (2) Technical contributions. (i) MoE with decoupled training addresses the limited modeling capacity and optimization difficulty of standard NCDE's dynamic function, which is a novel solution in irregular TS modeling. (ii) Using diffusion models to parameterize MoE-NCDE solves its generalization limitation to unseen samples, enabling effective continuous generation. These are novel methodological contributions rather than simple applications.
>
>
> **W2: The rationale for using dense MoE but not sparse MoE.**
>
> **A2:**
>
> We thank the reviewer for the insightful question. In fact, we have carefully considered this.
> Sparse MoE trains many experts but activates few during inference. Under high data missingness, insufficient data may cause expert collapse (please refer to our response to W2 of Reviewer WdCy). We added sparse vs. dense MoE comparisons (Please refer to **"Dense Best'' and "Sparse Best'' in Tables 1 and 2 in our response to Reviewer WdCy**). Results show that Dense MoE outperforms overall because: (i) activating all experts mitigates individual expert collapse, while sparse MoE may activate collapsed ones; (ii) dense MoE provides more stable gradients under missingness and requires training far fewer experts. This is the rationale for using dense MoE.
>
>
> **W3: Expert (MoE) diversity is not ensured and may learn redundancy.**
>
> **A3:**
>
> Thanks for the insightful observation. Actually, we have demonstrated MoE diversity: Quantitatively, Table 5 shows MoE significantly outperforms a single expert under the same parameters, and qualitatively, visualization (Figure 6) confirms experts capture distinct temporal patterns. However, as the reviewer noted, Figure 7 shows limited gains from increasing experts, due to: (1) sparse data patterns, the three datasets have only 2, 5, and 2 classes respectively, insufficient for more experts to find differentiated patterns; (2) high data missingness makes temporal patterns ambiguous, limiting effective training of additional experts even in dense MoE. We agree that exploring explicit diversity mechanisms for MoE is worthwhile, especially in scenarios with irregular missing data.   **We have added a limitations section in the revised manuscript and incorporated the above analysis into it**, e.g., sparse MoE training and expert diversity challenges  under missing and pattern sparse data.
>
> **Questions**
>
> **Q1: Why Single-expert model with 8 layers in Table 8 is worse than that with 2 layers in Figure 7?**
>
> **A1:**
>
> Thanks  for the valuable observations. This is actually a typo, the y-axes of the last two subplots in Figure 7 have different ranges from the first but were occluded. The corrected figure is provided in Link https://anonymous.4open.science/r/MN-Diff-Rebuttal/hyperana_mdd_fix_layout.jpg. We have fixed this in the paper. In fact, 8-layer single expert is not always worse than 2-layer (**Table 1** here): it underperforms on ECG200 and ECGFD due to overfitting (smaller data size and fewer class patterns), but outperforms on ECG5K which has more data and class diversity.
>
>
> **Table 1:  Comparison between single-expert model with 8 and 2 linear layers.**
> |    | ECG200-30%|ECG5K-30%| ECGFD-30%|ECG200-50%|ECG5K-50%| ECGFD-50%|ECG200-70%|ECG5K-70%| ECGFD-70%
> |:-:|:-:|:-:|:-:|:----:|:-:|:-:|:-:|:----:|:-:|
> |1-expert-8|0.288| **0.12**| **0.694** |0.321| **0.152**| 0.831| 0.314| **0.165**| 0.837|
> |1-expert-2|0.226 |0.152 |0.74 |0.255 |0.179 |0.778 |0.244 |0.196 |0.784|
>
>
>
>
> **Q2: How does the model scale on longer sequences 128, 256?**
>
> **A2:**
>
> We agree that model scale on longer sequences is important. Beyond short lengths (12, 24, 36), the four ECG datasets also include longer sequences (96, 140, 136, and 82; see Appendix D.1), where our method shows clear advantages. We further add experiments with longer sequence 256 across multiple datasets (**Table 2**), showing better performance. This demonstrates  generalizability.
>
>
> **Table 2:  TSG with sequence length 256 under various datasets. Due to limited space, the metric is first averaged on the  MDD and DS (Discriminative score) and then averaged on setting of 30%, 50%, 70% missing rates.**
> |    | Sines-30%-50%-70%| Stocks-30%-50%-70%| Energy-30%-50%-70%| MuJoCo-30%-50%-70%|
> |:-:|:-:|:-:|:-:|:-:|
> |Ours| **0.246**| **0.222**| **0.7**| **0.464**|
> |KoVAE| 0.69| 0.702| 0.915| 0.743|
> |GTGAN| 0.513| 0.469| 0.72| 0.682|
> |TimeGAN-NCDE| 1.041| 0.87| 0.972| 0.936|
> |TimeVAE-NCDE| 0.496| 0.495| 0.708| 0.488|
> |Diffusion-NCDE| 0.335| 0.396| 0.728| 0.538|
> |ProFITi| 0.584| 0.482| 0.704| 0.64|
> |HeTVAE| 1.134| 0.663| 1.001| 1.134|

---

> > ### Author Rebuttal · Reviewer_4pP9 · 2026-04-01
> >
> > Our concerns have been addressed.
> > Although some of the contributions still appear to be a combination of existing paradigms with limited theoretical innovation, the work offers a valuable exploration of new scenarios with empirical significance.
> > Therefore, we have raised the score.

---

> > > ### Author Response · Authors · 2026-04-01
> > >
> > > We are pleased that the reviewer’s concerns have been adequately addressed and appreciate the recognition of the empirical significance of our work in exploring new scenarios.
> > >
> > > We are also grateful for the careful evaluation and constructive feedback, which have significantly improved the manuscript.
> > >
> > > We further appreciate the reviewer’s positive reassessment and the decision to raise the score after the rebuttal.

---

### Official Review · Reviewer_f6Kp · 2026-03-08

**Soundness:** 3
**Presentation:** 3
**Significance:** 3
**Originality:** 3
**Overall Recommendation:** 5
**Confidence:** 4

**Summary:**

The paper proposes MN-Diff, a novel generative framework for continuous Time Series Generation (TSG) from irregularly sampled observations. Recognizing that existing Neural Controlled Differential Equation (NCDE) methods rely on a single dynamics function and tightly coupled optimization, the authors introduce three main contributions. First, they replace the single dynamics function in NCDEs with a Mixture-of-Experts (MoE) architecture to better capture complex temporal patterns. Second, they decouple the optimization process by using a frozen, pre-trained channel-wise autoencoder, allowing the training to focus specifically on the MoE dynamics. Finally, they address the issue of generating new samples by jointly training a diffusion model on the time series data and the learned MoE weights, enabling the generation of sample-specific parameters for new continuous sequences. The method is evaluated on ten datasets (four public, four medical, and two synthetic) and demonstrates superior performance compared to existing baselines.

**Compliance With Llm Reviewing Policy:**

Affirmed.

**Final Justification:**

The authors addressed my questions, I maintain my score.

**Key Questions For Authors:**

N/A

**Limitations:**

No, the authors have not adequately discussed the limitations and potential negative societal impact.

Suggestions for improvement: The current Impact Statement is practically non-existent, stating: "There are many potential societal consequences of our work, none of which we feel must be specifically highlighted here". The authors can replace this with a genuine discussion. For example, generating synthetic medical data (like the ECG datasets used ) carries risks if the generative model hallucinates features or inadvertently leaks patient privacy (even though MIR was evaluated, the broader risk remains). Additionally, the authors should include an explicit "Limitations" section in the main text discussing failure modes, the computational overhead mentioned in Section 4.5 , and the reliance on the quality of the cubic spline interpolation.

**Strengths And Weaknesses:**

**Strengths:**
- The submission is technically sound and supported by rigorous empirical evidence. The authors stress-test their model across ten diverse datasets --- four public, four medical, and two synthetic —under varying missing data rates of 30%, 50%, and 70%.
- The ablation studies effectively isolate and validate the contributions of the MoE functions and the decoupled optimization design. Furthermore, the authors provide solid theoretical and experimental justifications in the appendix for why NCDEs are better suited for this interpolation and continuous generation task than standard Neural ODEs.
- The paper is generally well-structured and logically flows from the limitations of current NCDEs to the proposed decoupled MoE solution.
- The work addresses a highly relevant and practical problem. Real-world data, particularly in domains like healthcare (e.g., Electronic Health Records), is often sparse and irregularly sampled, yet downstream tasks require dense, continuous data.

**Weaknesses:**
- The proposed MoE-NCDE inherently carries a higher time complexity than a standard NCDE. As the authors explicitly note, it takes 1.2s compared to 0.81s for a specific double-length generation test. This could be a bottleneck for very large-scale applications, though the authors partially mitigate this during inference by avoiding NCDE retraining via the diffusion parameterization.
- The authors dismissively state: "There are many potential societal consequences of our work, none of which we feel must be specifically highlighted here". This is inadequate for modern machine learning conferences, especially given that generating synthetic medical data (like the ECG datasets used) carries inherent risks regarding hallucination and privacy.

---

> ### Author Rebuttal · Authors · 2026-03-31
>
> **Weakness**
>
> **W1: MoE-NCDE inherently carries a higher time complexity than a standard NCDE and may have a bottleneck for very large-scale applications.**
>
> **A1:**
>
> We thank the reviewer for the insightful comment. Applying MoE-NCDE to large-scale data may indeed pose efficiency challenges. However, in practice, we notice that the overhead introduced by MoE is moderate and the primary bottleneck lies in the native NCDE mechanism itself. Therefore, developing architectures that significantly improve the native NCDE efficiency is a valuable future research direction.
>
> **W2: Regarding the revision of the impact statement.**
>
> **A2:**
>
> We thank the reviewer for the constructive suggestions. We have provided a more substantive discussion and rewritten the impact statement accordingly:
>
> This paper advances continuous time series generation applicable to diverse domains. While synthetic data generation offers benefits such as data augmentation and privacy-preserving sharing, it carries risks. For example, in healthcare, generated signals may contain hallucinated features that could mislead analysis if used without expert validation. Additionally, generative models may inadvertently memorize sensitive patterns from training data. We recommend applying appropriate privacy safeguards (e.g., differential privacy) before deploying on sensitive data, and caution that generated data should complement rather than replace real data in safety-critical applications.
>
> **Limitations**
>
> **L1: Regarding the improvement of the impact statement.**
>
> **A1:**
>
> We have provided a more genuine discussion of the potential impacts of our work. Please refer to our response to W2.
>
> **L2: The authors should include an explicit "Limitations" section in the main text discussing failure modes, the computational overhead mentioned in Section 4.5 , and the reliance on the quality of the cubic spline interpolation.**
>
> **A2:**
>
> We thank the reviewer for the valuable suggestion. We have added a "Limitations" section discussing:
>
> (1) MoE failure modes. We added dense vs. sparse MoE comparisons (**Table 1 to 2**). Both exhibit that mode may collapse as expert count increases, because data missingness and sparse patterns cannot support training more experts. Dense MoE outperforms sparse MoE  (please refer to **"Dense Best" vs. "Sparse Best" in Table 1 to 2**), because it typically has a smaller number of experts than Sparse-MoE and all experts participate in computation, providing more stable gradients and mitigating individual expert collapse, whereas sparse MoE may activate collapsed experts, leading to unstable performance. Training sparse MoE under data missingness and pattern sparsity remains challenging, and this work does not yet explore effective training architectures or mechanisms for this setting.
>
> (2) Scalability and data dependency: We incorporated the efficiency limitations of applying MoE-NCDE to large-scale data (as discussed in W1), and noted that MoE-NCDE's performance depends on the quality of available observations and interpolation functions.
>
>
>
>  **Table 1:  Expert collapse mode analysis on DS (Discriminative score). The metric is averaged on the setting of 30%, 50%, 70% missing rates with sequence length 24. For sparse MoE, we set a total of 8 experts, activating 1, 2, 4, or 6 experts, respectively.**
> |    | ECG200-30%-50%-70%| ECG5K-30%-50%-70%| ECGFD-30%-50%-70%|
> |:-:|:-:|:-:|:-:|
> |Dense-1-Expert| 0.268| 0.388| **0.13**|
> |Dense-2-Experts| **0.229**| 0.365| 0.153|
> |Dense-4-Experts| 0.315| **0.269**| 0.153|
> |Dense-6-Experts| 0.279| 0.287| 0.17|
> |------|------|------|------|
> |Sparse-1-Expert| 0.335| 0.292| 0.177|
> |Sparse-2-Experts| 0.371| **0.287**| **0.133**|
> |Sparse-4-Experts| **0.288**| 0.291| 0.173|
> |Sparse-6-Experts| **0.288**| 0.314| 0.143|
> |**Dense Best**| **0.229** (2 experts)| **0.269** (4 experts)| **0.13** (1 experts)|
> |**Sparse Best**| 0.288 (4 experts)| 0.287 (2 experts)| 0.133 (2 experts)|
> |**Win Count**| Dense: 7 | Sparse: 5 | Tie: 0 |
>
>
> **Table 2:  Expert collapse mode analysis on the MDD metric. Other settings are same as Table 1.**
> |    | ECG200-30%-50%-70%| ECG5K-30%-50%-70%| ECGFD-30%-50%-70%|
> |:-:|:-:|:-:|:-:|
> |Dense-1-Expert| 0.242| 0.176| 0.767|
> |Dense-2-Experts| **0.23**| 0.103| 0.541|
> |Dense-4-Experts| 0.231| 0.079| **0.518**|
> |Dense-6-Experts| 0.238| **0.077**| 0.542|
> |------|------|------|------|
> |Sparse-1-Expert| 0.235| 0.085| **0.542**|
> |Sparse-2-Experts| 0.248| 0.087| 0.595|
> |Sparse-4-Experts| 0.231| **0.082**| 0.568|
> |Sparse-6-Experts| **0.226**| 0.084| 0.551|
> |**Dense Best**| 0.23 (2 experts)| **0.077** (6 experts)| **0.518** (4 experts)|
> |**Sparse Best**| **0.226** (6 experts)| 0.082 (4 experts)| 0.542 (1 experts)|
> |**Win Count**| Dense: 6 | Sparse: 5 | Tie: 1 |

---

> > ### Author Rebuttal · Reviewer_f6Kp · 2026-04-01
> >
> > Thank you. I will maintain my recommendation for acceptance.

---

> > > ### Author Response · Authors · 2026-04-01
> > >
> > > We sincerely thank the reviewer for the acceptance recommendation and continued support.
> > >
> > > We greatly appreciate the reviewer’s time and constructive feedback throughout the review process, which has significantly  helped improve our manuscript.

---

### Official Review · Reviewer_WdCy · 2026-03-10

**Soundness:** 3
**Presentation:** 3
**Significance:** 3
**Originality:** 3
**Overall Recommendation:** 5
**Confidence:** 5

**Summary:**

This paper proposes MN-Diff, a framework for learning continuous time-series dynamics from irregular and sparse time series by combining a diffusion model with a mixture-of-experts (MoE) neural controlled differential equation (NCDE). The key idea is to use diffusion to generate both trajectories and sample-specific dynamics parameters, then use the continuous-time MoE-NCDE to refine them into realistic trajectories. The overall motivations and proposed methodology are clear and reasonable.

**Compliance With Llm Reviewing Policy:**

Affirmed.

**Final Justification:**

The concerns are cleared, while the paper does not reach the acceptance level.

**Key Questions For Authors:**

The main questions are same as weakness. For instance, why we do not use real world irregular benchmarks, like MIMIC and eICU, which are standard practices in related works. The authors only need to address weakness not here.

**Limitations:**

Same as above.

**Strengths And Weaknesses:**

Strengths:

1. The modeling of continuous trajectories for irregular, sparse observations is a critical need, particularly for clinical EHR data as the patient records are not routinely documented.

2. Integrating the NCDE into an MoE-NCDE framework with a diffusion-based model is creative. It provides a viable way to jointly model the irregular time series and the MoE weights.

3. The MoE provides a modeling of diverse trajectories from irregular data, which is important in clinical settings. For instance, a patient typically have various phenotypes, each may correspoding a different trajectories.

4. It replaces the State Initialization Network (SIN) and Readout Network (RN) with a pre-trained channel-wise autoencoder, making the model focus on the dynamics.

5. The experiments covers ECGs, forecasting, and synthetic cubic-polynomial recovery, providing a comprehensive evaluation of distributional quality and functional recovery in continuous spaces.

6. Strong and comprehensive analyses beyond the numerical results, such as Qualitative evaluation and Privacy analysis.

Weakness:

1. A key weakness for this paper is its extremely limited investigation of related works (only one paragraph).

Since prior work has already addressed irregular-to-regular prediction (fig 1.b) or arbitrary-time inference (fig 1.c), and in some cases both (e.g., TrajGPT [1]). The paper would be stronger with a more comprehensive discussion of different methods in these forecasting setups in Sec 2.

[1] TrajGPT: Irregular Time-Series Representation Learning of Health Trajectory. IEEE JBHI, 2025.

2. From my understanding, this paper does not explicitly analyze or address expert collapse or sample diversity issues. As this paper proposes a heavy-engineered and complex architecture, we may need investigate hyperparameter analysis on the number of experts and evaluations under different missing rates. More sensitivity and robustness analyses may needed to study expert collapse, collapse behavior, and sample diversity,

3. The Intro claims the irregularly-sampled time series modeling is cruical in applications like healthcare (ICU / EHR data). However, it lacks experiments on widely used ICU / EHR datasets, such as MIMIC and eICU datasets.

It seems that this paper focuses on the continous data with dropped time stamps (like 30%) instead of real and widely used irregular data such as MIMIC / eICU datasets. Without these real benchmarks, they are hard to compare with existing methods like neural ODE/CDE family or HeTVAE.

---

> ### Author Rebuttal · Authors · 2026-03-31
>
> **Weakness**
>
> **W1: Limited investigation of related works.**
>
> **A1:**
>
> Except for main-text paragraph, we also include additional discussion like "TS modeling with irregular data" in the Appendix A.1.
> We thank the reviewer for pointing out more relevant work here. We have included TrajGPT in the related work part and expanded the discussion on methods for modeling irregular data. Due to limited space,  the summary is as follows:
>
> Although limited work directly addresses continuous/irregular TSG, several studies explore related problems: GRU-D and ODE-RNN extend RNNs with temporal decay and ODE-based hidden state evolution; HeTVAE and STraTS leverage attention mechanisms for irregular intervals; ProFITi  applies Conditional Normalizing Flows for irregular TS forecasting; TrajGPT introduces Selective Recurrent Attention for irregular event prediction.
>
> They provide more context for understanding MN-Diff and inspire further research on continuous and irregular TSG.
>
>
> **W2: Lack analysis on expert collapse, sample diversity, and sensitivity analyses on the number of experts with missing rates.**
>
> **A2:**
>
> (1) Expert number sensitivity. Actually, the original paper already includes sensitivity analysis across different missing rates (30%/50%/70%) on three medical datasets, as shown in Figure 7 and Appendix Figure 8.
>
> (2) Expert collapse. Thanks for the valuable insight. We have added a detailed analysis of expert collapse and included additional sparse MoE experiment to better understand it. Both dense and sparse MoE exhibit performance degradation as expert count increases, indicating expert collapse, as shown in **Table 1 to 2**. The causes include:
>
> (i) data missingness limits effective training of more experts;
>
> (ii) datasets with few classes (e.g., 2-class ECG200/ECGFD) lack sufficient patterns for many experts;
>
> (iii) dense MoE collapses less severely than sparse MoE ("Dense Best" is better than "Sparse Best" in the **Tables 1 and 2**), as sparse gating typically involves a larger number of experts and data with missingness doesn't support sparse expert specialization training.
>
> (3) Sample diversity. In paper, MDD measures distribution fidelity while DS  (Discriminative score) evaluates diversity. Hence, we have measured sample diversity and MN-Diff achieves superior DS compared to baselines. Besides, Dense MoE shows better diversity than sparse MoE due to easier training under data missingness, e.g., "Dense Best" is better than "Sparse Best" in the **Tables 1 and 2**.
>
> **We have added a limitations section in the revised manuscript and incorporated the above analysis into it**, e.g., sparse MoE training and expert specialization challenges under data with missingness and sparse pattern.
>
>
> **Table 1:  Expert collapse mode analysis on DS. The metric is averaged on the setting of 30%, 50%, 70% missing rates with sequence length 24. For sparse MoE, we set a total of 8 experts, activating 1, 2, 4, or 6 experts, respectively.**
> |    | ECG200-30%-50%-70%| ECG5K-30%-50%-70%| ECGFD-30%-50%-70%|
> |:-:|:-:|:-:|:-:|
> |Dense-1-Expert| 0.268| 0.388| **0.13**|
> |Dense-2-Experts| **0.229**| 0.365| 0.153|
> |Dense-4-Experts| 0.315| **0.269**| 0.153|
> |Dense-6-Experts| 0.279| 0.287| 0.17|
> |------|------|------|------|
> |Sparse-1-Expert| 0.335| 0.292| 0.177|
> |Sparse-2-Experts| 0.371| **0.287**| **0.133**|
> |Sparse-4-Experts| **0.288**| 0.291| 0.173|
> |Sparse-6-Experts| **0.288**| 0.314| 0.143|
> |**Dense Best**| **0.229** (2 experts)| **0.269** (4 experts)| **0.13** (1 experts)|
> |**Sparse Best**| 0.288 (4 experts)| 0.287 (2 experts)| 0.133 (2 experts)|
> |**Win Count**| Dense: 7 | Sparse: 5 | Tie: 0 |
>
>
> **Table 2:  Expert collapse mode analysis on MDD.**
> |    | ECG200-30%-50%-70%| ECG5K-30%-50%-70%| ECGFD-30%-50%-70%|
> |:-:|:-:|:-:|:-:|
> |Dense-1-Expert| 0.242| 0.176| 0.767|
> |Dense-2-Experts| **0.23**| 0.103| 0.541|
> |Dense-4-Experts| 0.231| 0.079| **0.518**|
> |Dense-6-Experts| 0.238| **0.077**| 0.542|
> |------|------|------|------|
> |Sparse-1-Expert| 0.235| 0.085| **0.542**|
> |Sparse-2-Experts| 0.248| 0.087| 0.595|
> |Sparse-4-Experts| 0.231| **0.082**| 0.568|
> |Sparse-6-Experts| **0.226**| 0.084| 0.551|
> |**Dense Best**| 0.23 (2 experts)| **0.077** (6 experts)| **0.518** (4 experts)|
> |**Sparse Best**| **0.226** (6 experts)| 0.082 (4 experts)| 0.542 (1 experts)|
> |**Win Count**| Dense: 6 | Sparse: 5 | Tie: 1 |
>
>
> **W3: Lack ICU/EHR datasets with inherently missing values.**
>
> **A3:**
>
> We thank the reviewer for the valuable insight. Our task focuses on continuous TSG, and although MN-Diff can be applied to naturally irregular data to generate values at naturally missing time steps, evaluation is challenging because these values doesn't have ground truth.
>
> Hence, following prior works such as KOVAE and GTGAN, we simulate missingness by randomly dropping 30%/50%/70% of regular data and use the held-out values as ground truth for evaluation.
>
> Developing evaluation protocols for naturally irregular data remains an important future direction.

---

> > ### Author Rebuttal · Reviewer_WdCy · 2026-04-01
> >
> > It solves my concerns.

---

> > > ### Author Response · Authors · 2026-04-01
> > >
> > > We sincerely thank the reviewer for the time and effort devoted to the review process, which has greatly helped improve the quality of our manuscript.
> > >
> > > We are pleased that the reviewer's concerns have been adequately addressed, and sincerely appreciate the reviewer’s decision to raise the score after rebuttal.

---

### Decision · Program_Chairs · 2026-04-30

**Decision:**

Accept (regular)

**Comment:**

This paper focuses on generating time series regardless of sampling intervals and skips. The problem setting is one of the most general settings for time series generation as I know. To this end, the authors combine various state-of-the-art design concepts, NCDE, MoE, and diffusion. It looks quite complicated to integrate them in a smooth manner and they accomplished good performance in comparison with other baselines.

Reviewers raised several questions and but the authors successfully addressed them via additional experiments. Since the model architecture is complicated, it is important to understand the contribution by each design point.

Overall, this paper presents timely contributions for time series generation. One minor point that I am aware of is that there are more baselines based on diffusion in this topic, and it is beneficial to readers if they conduct more experiments with diffusion-based time series generation methods.